# ReDiG: Reinforced Diffusion on Graphs for Decentralized Coordinated Multi-Robot Navigation with Smooth Formation Adaptation

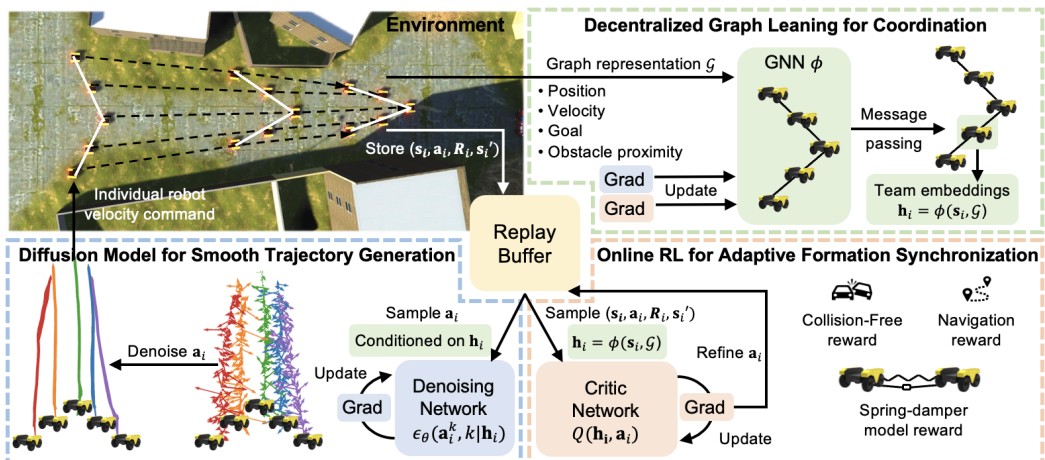

Figure 1: ReDiG enables decentralized multi-robot navigation with smooth formation adaptation by integrating decentralized graph learning for coordination, diffusion models for smooth trajectory generation, and online reinforcement learning for formation synchronization.

## Abstract

Coordinated navigation is a fundamental capability for multi-robot teams to traverse complex unstructured environments. During navigation, robots are often required to maintain mission-specific formations, such as wedge formations for enhanced visibility and area coverage. However, rigid formations can hinder navigation in challenging scenarios like narrow corridors, which demand formation adaptation. Reinforcement learning (RL) is commonly used for coordinated multi-robot navigation due to its ability to learn through interaction with the environment. However, its step-wise decision-making process often results in jerky motion. In contrast, diffusion models generate smoother trajectories through probabilistic denoising, but rely heavily on high-quality demonstrations. Collecting such demonstrations is challenging in multi-robot systems due to the coordination and synchronization required among individual robots. To address these issues, we introduce a novel method named *Reinforced Diffusion on Graphs* (ReDiG) to enable decentralized coordinated multi-robot navigation with smooth formation adaptation. Under a unified learning paradigm, ReDiG integrates: (1) graph learning for decentralized coordination to enable formation adaptation, (2) diffusion models for generating smooth individual robot trajectories, and (3) online RL to refine noisy demonstrations through leveraging feedback from environment interaction, which enables robot synchronization and guides effective diffusion training. We evaluate ReDiG through extensive experiments in both indoor and outdoor environments using physical robot teams and robotics simulations. Experimental results show that ReDiG enables smooth formation adaptation and achieves state-of-the-art performance in coordinated multi-robot navigation within complex environments. More details are available on the project website: https://anonymous23885.github.io/ReDiG.

# 1 INTRODUCTION

Multi-robot systems have gained significant attention in recent years due to their advantages, such as redundancy Gao et al. (2023), operational efficiency Rekleitis et al. (2008), and scalability Balch & Hybinette (2000), which make them well-suited for a wide range of real-world large-scale applications, including search and rescue Queralta et al. (2020); Yang & Parasuraman (2020); Baxter et al. (2007), transportation Amanatiadis et al. (2015); Koung et al. (2021); Farivarnejad et al. (2021), and space exploration Han et al. (2020); Indelman (2018); Martinez Rocamora Jr et al. (2023). Coordinated multi-robot navigation is an essential capability, which enables teams of robots to traverse environments in a synchronized manner. During coordinated navigation, robot teams are typically required to maintain task-specific formations, such as a wedge formation, to enhance visibility and area coverage. To support this, multi-robot coordination is critical, which enables robots to share information with their teammates, particularly in a decentralized manner. However, rigid formations can hinder progress and impede navigation in complex environments, such as narrow corridors. To overcome this, multi-robot synchronization is essential, allowing robots to align their actions in both time and space to maintain and adapt formations. Furthermore, such constrained scenarios often require frequent motion adjustments for formation adaptation, which can lead to non-smooth and inefficient traversal, which results in an additional challenge that must be addressed.

Due to the importance of coordinated multi-robot navigation, a wide range of methods have been developed. Traditional formation control methods, including leader-follower Wu et al. (2022) and virtual region methods Abujabal et al. (2023), often rely on preset rigid formation shapes. However, such formations lack the flexibility to adapt to complex environments. Learning-based methods, such as reinforcement learning (RL) Hu et al. (2023), address this limitation by optimizing robot actions through interaction with the environment. However, due to the stepwise nature of RL decision-making, robots stop or adjust their motion frequently in pursuit of higher rewards, resulting in jerky trajectories. Generative models such as diffusion models have recently shown promise in offline RL Zhu et al. (2024) by leveraging reward signals from fixed datasets and iteratively denoising demonstrations to generate smooth trajectories for robot navigation. However, collecting expert demonstrations in multi-robot systems is challenging due to the need for precise coordination and synchronization among individual robots. To overcome this, recent studies explore diffusion models in online RL, refining actions through gradient ascent in an off-policy setting Yang et al. (2023). However, none of the existing online diffusion methods have been applied to multi-robot systems.

To address the above limitations in the current state of the art, we introduce the first online diffusion-based multi-robot learning approach called *Reinforced Diffusion on Graph* (**ReDiG**), to enable a new multi-robot capability of decentralized coordinated multi-robot navigation with smooth formation adaptation. Specifically, ReDiG represents a robot team as a graph, where each node represents a robot along with its attributes such as position, velocity, goal, and obstacle proximity, and each edge encodes the spatial relationship between robot pairs. ReDiG integrates three learning components into a unified approach to enable coordinated multi-robot navigation with formation adaptation: First, a decentralized graph neural network computes team-level embeddings from the graph representation, which captures the team context for effective coordination. Second, a diffusion model on each robot learns a navigation policy that generates smooth trajectories conditioned on the team embedding. Third, an online RL module synchronizes individual robot actions through iterative interaction with the environment, enabling formation adaptation in dynamic and constrained scenarios.

Our primary contribution is the introduction of the ReDiG approach to enable coordinated multi-robot navigation with smooth formation adaptation. The specific novelties include:

- From the perspective of robot capability, we develop one of the first learning-based solutions for decentralized coordinated multi-robot navigation with formation adaptation. ReDiG not only enables a new capability of coordinated navigation with formation adaptation, but also improves motion trajectory smoothness for individual robots, particularly when traversing narrow corridors that require frequent adjustments.

- From the perspective of algorithmic novelty, we introduce ReDiG as the first online diffusion-based multi-robot learning paradigm, which unifies decentralized graph learning for team-level coordination, diffusion models for smooth individual trajectory generation, and online reinforcement learning to iteratively refine noisy demonstration, ensuring coordinated and synchronized multi-robot adaptive formation control across the robot team.

## 2 RELATED WORK

**Coordinated Multi-Robot Navigation.** Traditional coordinated navigation with formation control often relies on manually designed strategies, such as the leader-follower Reily et al. (2020); Wu et al. (2022), and Virtual region methods Abujabal et al. (2023); Alonso-Mora et al. (2019) However, these formations are often rigid and fail to traverse constrained space. Learning-based methods introduce greater flexibility: Graph Neural Networks (GNNs) improve team coordination in a decentralized manner Goarin & Loianno (2024); Gao et al. (2024), and Reinforcement Learning (RL) utilizes reward signals to enable robot teams to learn coordinated behaviors that are difficult to manually design Blumenkamp et al. (2022); Hu et al. (2023). However, in multi-robot navigation with synchronization, RL often suffers from step-wise decision-making, which causes robots to stop or adjust their motion frequently to maximize immediate rewards, leading to jerky trajectories and reduced smoothness.

**Diffusion Models for Robot Policy Learning.** Diffusion models have recently been applied in robotics to generate smooth trajectories through iterative denoising. For single-robot planning, diffusion models are used to sample motion plans conditioned on environmental context Fang et al. (2024); Xian & Gkanatsios (2023). For multi-robot planning, Motion Diffuser Jiang et al. (2023) enables trajectory prediction for multi-robot through cost function, Resilient Distributed Diffusion Li et al. (2020a) enables resilient distributed control under adversarial conditions based on the centerpoint concept, MMD Shaoul et al. (2024) generates collision-free multi-robot trajectories based on single-robot data. However, applying diffusion models to multi-robot systems remains challenging due to the need for large-scale, well-synchronized expert demonstrations, which are difficult to obtain.

**Diffusion for Offline RL.** Diffusion models have been recently integrated with offline RL to improve policy through generative sampling guided by RL signals. Diffusion-QL Wang et al. (2022) biases diffusion sampling toward high-value actions using Q-learning. SRDP Ada et al. (2024) enhances out-of-distribution (OOD) generalization by reconstructing state representations. Diffuser Janner et al. (2022) applies reward signals at the trajectory level, while Simple Hierarchical Chen et al. (2024) extends this to multi-task settings using hierarchical diffusion policies. MTDiff He et al. (2023) further supports multi-task planning through transformer-based conditioning. MADiff Zhu et al. (2024) is the first offline diffusion-based multi-agent framework. However, for complex behaviors that demand coordination and synchronization, which are rarely available in offline datasets, offline RL struggles to learn behaviors that are absent from expert demonstrations.

**Diffusion for Online RL.** Diffusion-based online RL addresses the limitation of offline RL through directly interacting with the environment, enabling the model to explore and refine behaviors beyond those available in expert demonstrations. DIPO Yang et al. (2023) is the first to integrate diffusion policies into online RL and introduces a novel diffusion policy improvement method, which uses off-policy to refine actions through gradient ascent updates to obtain higher rewards. QSM Psenka et al. (2023) aligns the diffusion model's score function with the gradient of a Q-function, which enables efficient policy updates. QVPO Ding et al. (2024) introduces a Q-weighted variational loss to ensure robust policy improvement with enhanced exploration. However, none of these diffusion-based online RL methods have been applied to multi-robot systems, particularly those requiring multi-robot coordination and synchronization.

## 3 APPROACH

### 3.1 PROBLEM DEFINITION

We represent a team of $n$ robots as an undirected graph $\mathcal{G} = \{\mathcal{V}, \mathbf{E}\}$. Each robot is represented as a node within the node set $\mathcal{V} = \{\boldsymbol{v}_1, \boldsymbol{v}_2, \ldots, \boldsymbol{v}_n\}$. The attributes of each robot $i$ are represented by $\boldsymbol{v}_i = [\mathbf{p}_i, \mathbf{q}_i, \mathbf{g}_i]$, where $\mathbf{p}_i = [p_i^x, p_i^y]$ denotes its position, $\mathbf{g}_i = [g_i^x, g_i^y]$ represents its goal position, and $\mathbf{q}_i = [q_i^x, q_i^y]$ defines its velocities along the x and y directions. The edge matrix $\mathbf{E} = \{a_{i,j}\}^{n \times n}$ represents the spatial adjacency between robots, where $a_{i,j} = 1$, if the $i$-th robot and the $j$-th robot are within a radius; otherwise $a_{i,j} = 0$. We define the state of the $i$-th robot as the concatenation of its attributes, $\mathbf{s}_i = [\mathbf{p}_i, \mathbf{q}_i, \mathbf{g}_i, d_i]$, where $d_i$ represents the distance from the nearest obstacle to the $i$-th robot. We further define the action of the $i$-th robot as $\mathbf{a}_i = [v_i^x, v_i^y]$, where $v_i^x$ and $v_i^y$ represent the output velocities in the $x$ and $y$ directions, respectively.

We address decentralized coordinated multi-robot navigation, while ensuring both smooth individual robot motion and adaptive formation control for the entire robot team. Specifically, we aim to address:

- **Decentralized Coordination** enables each robot to coordinate with teammates by sharing and integrating state information in a fully decentralized manner, ensuring coherent multi-robot coordination without centralized control.
- **Smooth Trajectory Generation** enables each robot to generate smooth, collision-free trajectories that enable stable motion and efficient goal-reaching.
- **Adaptive Formation Synchronization** enables robots to align their motion in both time and space, thus maintaining mission-specific formations while dynamically adjusting their spatial team configuration to navigate through constrained environments.

## 3.2 Reinforced Diffusion on Graphs (ReDiG)

To enable decentralized coordinated multi-robot navigation with smooth formation adaptation, we propose the novel ReDiG approach with a unified learning paradigm, using a graph neural network to coordinate robots within the team in a decentralized way, a diffusion model to control individual robots for smooth navigation, and online reinforcement learning for adaptive formation synchronization. Our ReDiG approach is illustrated in Figure 1.

Given the robot team's graph representation $\mathcal{G}$ and robot states $\mathbf{s}_i$ of the $i$-th robot, we develop a graph neural network $\phi$ to encode spatial relationships among robots in the team and then compute the embedding $\mathbf{h}_i = \phi(\mathbf{s}_i, \mathcal{G})$ of the team state for each $i$-th robot. The graph network $\phi$ consists of linear layers that first map the robot state $\mathbf{s}_i$ to the individual embedding $\mathbf{z}_i$ of the $i$-th robot by $\mathbf{z}_i = \mathbf{W}^z \mathbf{s}_i$, where $\mathbf{W}^z$ denotes the learnable weight matrix of the linear layers. By using message passing, $\phi$ then aggregates $\mathbf{z}_i$ with the embeddings of all other teammates within a spatial radius to compute the final team state embedding $\mathbf{h}_i$ for the $i$-th robot, which is defined as $\mathbf{h}_i = \mathbf{W}^h \mathbf{z}_i + \sum_{j \in \mathcal{N}(i)} \mathbf{W}^h (\mathbf{z}_j - \mathbf{z}_i)$, where $\mathbf{W}^h$ is the learnable weight matrix. For each $i$-th robot, the first term captures its individual state, while the second term encodes its relative spatial relationships with teammates, representing the team-level context. The learnable weight matrix $\mathbf{W}^h$ enables each robot to determine which information from teammates is most critical for its decision-making. The graph neural network $\phi$ is agnostic to the number of robots, which is able to aggregate arbitrary embeddings from neighborhood robots within a spatial radius, thus enabling decentralized team-level context embedding.

### 3.2.1 Smooth Action Generation for Individual Robots

We design a diffusion model $\psi$ conditioned on the team-level graph embedding $\mathbf{h}_i$ to generate smooth actions $\mathbf{a}_i$ for each robot, while maintaining awareness of its teammates' states. Formally, we model the individual action probabilistically as a conditional distribution $p(\mathbf{a}_i|\mathbf{h}_i)$. However, due to the high dimensionality of the continuous action space, directly modeling or sampling from $p(\mathbf{a}_i|\mathbf{h}_i)$ is intractable. To address this, the diffusion model $\psi$ is developed to approximate the conditional distribution via a parameterized denoising process.

**Smooth Trajectory Generation.** The diffusion model is built upon a probabilistic diffusion process that consists of a forward and a reverse process. In the forward process, Gaussian noise is progressively added to the ground-truth action $\mathbf{a}_i^0$, which can be provided through expert demonstrations. A noise variance schedule $\{\beta^k\}_{k=1}^K$ with $\beta^k \in (0, 1)$ determines the variance of the noise added at each step $k$. This results in a sequence of noisy actions $\mathbf{a}_i^1, \mathbf{a}_i^2, \ldots, \mathbf{a}_i^k$. Formally, the forward process is defined as $q(\mathbf{a}_i^k|\mathbf{a}_i^{k-1}) = \mathcal{N}(\mathbf{a}_i^k; \sqrt{1-\beta^k}\mathbf{a}_i^{k-1}; \beta^k \mathbf{I})$. In the reverse process, the diffusion model $\psi$ iteratively reconstructs the ground-truth action $\mathbf{a}_i^0$. At each step $k$, the model learns to denoise $\mathbf{a}_i^k$ to recover $\mathbf{a}_i^{k-1}$, which progressively refines the trajectory toward the demonstrated behavior. This reverse process is governed by three key coefficients that jointly ensure smooth transitions across diffusion steps: $\lambda^k = 1/\sqrt{1-\beta^k}$ scales the denoised prediction, $\alpha^k = \prod_{i=1}^k (1-\beta^i)$ progressively reduces the influence of noise, and $\sigma^k$, derived from $\beta_k$, regulates exploration by controlling the amount of noise injected during sampling. To estimate the noise added during the forward process, we train a neural network $\epsilon_\theta(\mathbf{a}_i^k, k)$ parameterized by $\theta$. Then, the inverse process is defined as $\mathbf{a}_i^{k-1} = \lambda^k(\mathbf{a}_i^k - \alpha_k \epsilon_\theta(\mathbf{a}_i^k, k)) + \sigma^k z$, where $z \sim \mathcal{N}(0, \mathbf{I})$ denotes standard Gaussian noise. The

unstructured Gaussian noise introduces perturbations to the ground-truth action, which can result in jerky motion. The denoising network $\epsilon_\theta(\mathbf{a}_i^k, k)$ learns to remove these high-frequency perturbations, guided by three coefficients in the reverse process, thereby enabling smooth trajectory generation.

**Incorporating Team Context.** To enable team-aware trajectory generation for individual robots, we condition each robot's control on the team state embedding $\mathbf{h}_i$. Formally, this is expressed as: $\mathbf{a}_i^{k-1} = \lambda^k(\mathbf{a}_i^k - \alpha^k\epsilon_\theta(\mathbf{a}_i^k, k|\mathbf{h}_i)) + \sigma^k z$. We learn this conditioned reverse process by training the model to predict the actual noise $\epsilon_k$ added at the $k$-th step during the forward process, progressively recovering the ground-truth actions. The loss function is defined as follows:

$$\min_{\epsilon_\theta} \mathbb{E}_{\mathbf{h}_i, \mathbf{a}_i^0} \left[ \|\epsilon_k - \epsilon_\theta(\mathbf{a}_i^0 + \epsilon_k, k|\mathbf{h}_i)\|_2^2 \right] \tag{1}$$

The denoising network $\epsilon_\theta$ serves as the core learnable component of the diffusion model $\psi$, iteratively denoising a randomly sampled noise through a sequence of reverse diffusion steps conditioned on the team-level embedding to generate individual actions.

We theoretically analyze the upper bound of the loss defined in Eq. (1). Theorem 1 measures how well the learned action distribution $\hat{p}_\theta(\mathbf{a}_i^0|\mathbf{h}_i)$ approximates the true distribution $p(\mathbf{a}_i^0|\mathbf{h}_i)$. This upper bound has three components: 1). *Prior Mismatch* measures how closely the marginal distribution $q(\mathbf{a}_i^K)$ matches the Gaussian prior $p(\mathbf{a}_i^K) = \mathcal{N}(0, I)$. A poor match degrades initial samples, reducing trajectory feasibility. 2). *Matching Error* captures the cumulative noise prediction error across all diffusion steps. Inaccurate denoising leads to poor action reconstruction, affecting trajectory smoothness. 3). *Discretization Error* arises from approximating a continuous diffusion process with a finite number of steps, which may result in suboptimal trajectories. See Appendix C for proof.

**Theorem 1.** *Let $\hat{p}_\theta(\mathbf{a}_i^0|\mathbf{h}_i)$ be the learned action distribution and $p(\mathbf{a}_i^0|\mathbf{h}_i)$ the true distribution, with $K$ finite diffusion steps approximating the continuous process. The upper bound of the loss of the Conditional Diffusion Policy on Graph is bounded by the combination of prior mismatch, matching error, and discretization error:*

$$D_{\mathrm{KL}}(\hat{p}_\theta(\mathbf{a}_i^0|\mathbf{h}_i) \,\|\, p(\mathbf{a}_i^0|\mathbf{h}_i)) \leq \overbrace{D_{\mathrm{KL}}(q(\mathbf{a}_i^K) \,\|\, p(\mathbf{a}_i^K))}^{\text{Prior mismatch}} + \overbrace{\sum_{k=1}^{K} \mathbb{E}_{\mathbf{a}_i^0, \epsilon, k}\left[\|\epsilon - \epsilon_\theta(\mathbf{a}_i^k, k|\mathbf{h}_i)\|_2^2\right]}^{\text{Matching error}} + \overbrace{\mathcal{O}\left(\sum_{k=1}^{K}(\beta^k)^2\right)}^{\text{Discretization error}}$$

### 3.2.2 Adaptive Formation Synchronization

To maintain and adapt formations as robot teams navigate through constrained environments, it is essential to synchronize individual robot motions in both spatial and temporal dimensions. However, the diffusion model is trained using expert demonstrations, which are often imperfect and noisy (e.g., when they are provided by human experts or algorithms with access to privileged information). To address this limitation, we design a new online actor-critic RL framework that uses the unsynchronized actions generated by the diffusion model as initial guidance and refines the actions through reward-driven feedback, while simultaneously promoting formation-aware synchronization across the team.

**Synchronization for Formation Adaptation.** To enable multi-robot synchronization for adaptive formation control, we design a reward function inspired by the spring-damper model Deng et al. (2025a); Gabellieri et al. (2021). The spring component maintains a balance between keeping pairs of robots close enough to navigate through constrained environments (e.g., narrow corridors) and maintaining sufficient distance to avoid collisions, thus providing the flexibility necessary for adaptive formation control. The spring effect is modeled as $|d_{i,j} - p_{i,j}|$, where $d_{i,j}$ represents the expected distance in the original rigid formation and $p_{i,j}$ is the actual distance between the $i$-th and $j$-th robots, computed as $\|\mathbf{p}_i - \mathbf{p}_j\|_2$. The damper component mitigates oscillations and prevents overshooting by regulating the relative velocities between robot pairs, which is defined as $q_{i,j} = \|\mathbf{q}_i - \mathbf{q}_j\|_2$. By integrating both components, we formulate the spring-damper model as a reward function, defined as $R^{adp} = \sum_{\mathbf{v}_i, \mathbf{v}_j \in \mathcal{V}} -\omega|d_{i,j} - p_{i,j}| - (1 - \omega)q_{i,j}$, where $\omega$ is a hyperparameter that balances the contributions of the spring and damper components. The final reward $R = R^{adp} + R^{collision}$, where $R^{collision}$ denotes the obstacle avoidance reward for each individual robot Chen et al. (2017).

**Individual Action Refinement.** To refine the actions generated by the diffusion model $\psi$, we treat it as the actor network. A deep Q network $Q(\mathbf{h}_i, \mathbf{a}_i)$ is designed to serve as a critic network

to evaluate the actor. Using gradient ascent, these gradients are then applied to refine the actions generated by the diffusion model. The diffusion model $\psi$ subsequently treats the refined actions as updated ground-truth actions for further learning.

Formally, the critic network $Q(\mathbf{h}_i, \mathbf{a}_i)$ consists of an encoder followed by a MLP to generate the Q-value of a state-action pair $(\mathbf{h}_i, \mathbf{a}_i)$. The target value (i.e., TD target) of the critic network is computed using the Bellman equation by combining the immediate reward with a discounted estimate of future value, defined as $\mathbf{y}_i = R(\mathbf{s}_i, \mathbf{a}_i) + \gamma Q(\mathbf{h}'_i, \mathbf{a}'_i)$, where $\gamma$ is the discount factor, and $\mathbf{h}'_i$, $\mathbf{a}'_i$ denote the state embedding and action at the next time step for robot $i$. To train the critic network, we define the critic loss as $\mathbb{E}\big[(Q(\mathbf{h}_i, \mathbf{a}_i) - \mathbf{y}_i)^2\big]$, which minimizes the squared error between the predicted value and the target value. To stabilize training, we introduce two identical critic networks $Q_1(\mathbf{h}_i, \mathbf{a}_i)$ and $Q_2(\mathbf{h}_i, \mathbf{a}_i)$, and modify the loss function as follows:

$$\mathbb{E}\big[(Q_1(\mathbf{h}_i, \mathbf{a}_i) - \mathbf{y}_i)^2 + (Q_2(\mathbf{h}_i, \mathbf{a}_i) - \mathbf{y}_i)^2\big] \tag{2}$$

The use of two identical critic networks mitigates overestimation bias by encouraging consistent value estimates and reducing variance in target value predictions, which leads to stable learning.

We provide a theoretical analysis in Theorem 2 for the upper bound of the critic loss defined in Eq.(2), by deriving the approximation error between the learned value function $\hat{Q}^M$ and the true value function $Q^*$. This upper bound consists of two components: 1). *Statistical error* captures the limits of finite-sample estimation, amplified by replay buffer coverage mismatch. 2). *Algorithmic error* captures the approximation gap due to finite training iterations $M$. See Appendix C for proof.

**Theorem 2.** *Let $\hat{Q}^M$ be the learned value function after $M$ training steps and $Q^*$ be the true value function. Let $C$ be the constant denoting how well the replay buffer sampling covers the embedding-action space, $S$ be the constant denoting the worst-case single-step Bellman regression error, and $R_{max}$ be the maximum reward bound. The upper bound of the loss of the critic networks is bounded by the combination of statistical error and algorithmic error:*

$$\|\hat{Q}^M - Q^*\| \leq \overbrace{\frac{2\gamma C}{(1-\gamma)^2}S}^{\text{Statistical error}} + \overbrace{\frac{4\gamma^{M+1}}{(1-\gamma)^2}R_{\max}}^{\text{Algorithmic error}}$$

With the convergence guarantee from Theorem 2, the critic provides reliable Q-value estimates for actions generated by the diffusion model, which is also theoretically guaranteed to converge from Theorem 1. We can then refine actions to achieve higher values by computing the gradient of the minimum estimated value from $Q_1$ and $Q_2$ with respect to the action as $\nabla_{\mathbf{a}_i}Q(\mathbf{h}_i, \mathbf{a}_i)$, where $Q = \min(Q_1, Q_2)$. This gradient indicates the direction of the action $\mathbf{a}_i$ refinement, which increases its estimated value. The refined action is defined as $\mathbf{a}_i = \mathbf{a}_i + \eta\nabla_{\mathbf{a}_i}Q(\mathbf{h}_i, \mathbf{a}_i)$, where $\eta$ is the step size controlling how large to modify the action.

### 3.2.3 UNIFIED TRAINING AND DECENTRALIZED EXECUTION

**Unified Training of Graph, Diffusion, and RL Networks.** ReDiG includes three learning components, including a decentralized graph neural network $\phi$ for multi-robot coordination, a diffusion model $\psi$ for smooth individual trajectory generation, and an online RL to refine robot actions and enable adaptive formation control. To train all these components in a unified learning paradigm, ReDiG computes the gradients from both the diffusion loss in Eq. (1) and the critic loss in Eq. (2). Then, the gradients are backpropagated to update the graph network weight matrix $\mathbf{W}^z$ and $\mathbf{W}^h$, the denoising network $\epsilon_\theta$ in the diffusion model $\psi$, and the critic network $Q_1$ and $Q_2$. The unified training algorithm for ReDiG with a detailed explanation is presented in the Appendix D.

**Decentralized Execution.** ReDiG performs fully decentralized execution in multi-robot systems under a closed-loop control scheme. Each robot maintains its own 2D occupancy map of the environment and exchanges state information only with nearby teammates within a communication radius through wireless communication (e.g., Wi-Fi). At every timestep, each robot independently applies the trained graph neural network $\phi$, with shared weights $\mathbf{W}^z$ and $\mathbf{W}^h$, to process information from its neighbors and compute its local embedding $\mathbf{h}_i$. Finally, based on this local embedding $\mathbf{h}_i$, each robot runs its own instance of the trained diffusion policy $\psi$ to generate and execute velocity commands $\mathbf{a}_i$, thereby continuously adapting to its local observations in a closed-loop manner.

Table 1: Quantitative comparison of ReDiG and prior methods from Gazebo simulations in ROS2.

| Method | Circle Formation | | | | | Wedge Formation | | | | | Line Formation | | | | |
|---|---|---|---|---|---|---|---|---|---|---|---|---|---|---|---|
| | SR (%) | TT (sec) | $\delta < 0.5$ | $\delta < 0.1$ | $\delta < 0.03$ | SR (%) | TT (sec) | $\delta < 0.5$ | $\delta < 0.1$ | $\delta < 0.03$ | SR (%) | TT (sec) | $\delta < 0.5$ | $\delta < 0.1$ | $\delta < 0.03$ |
| L&F | 60.00 | 15.80 | 75.11 | 70.10 | 66.04 | 60.00 | 17.12 | 81.44 | 66.70 | 62.86 | 60.00 | 13.35 | 63.76 | 55.59 | 55.59 |
| DGNN | **100.00** | 34.70 | 60.41 | 58.91 | 58.91 | **100.00** | 63.70 | 47.85 | 42.33 | 41.92 | **100.00** | 36.80 | 27.90 | 20.16 | 20.16 |
| AFOR | **100.00** | 30.50 | **92.89** | **90.40** | **88.66** | **100.00** | 51.50 | **91.39** | **90.35** | **87.96** | **100.00** | 183.50 | 88.60 | 85.13 | 72.90 |
| ReDiG (ours) | **100.00** | **13.10** | 84.51 | 81.52 | 78.97 | **100.00** | **12.46** | 82.43 | 81.21 | 80.37 | **100.00** | **10.03** | **91.17** | **87.68** | **80.91** |

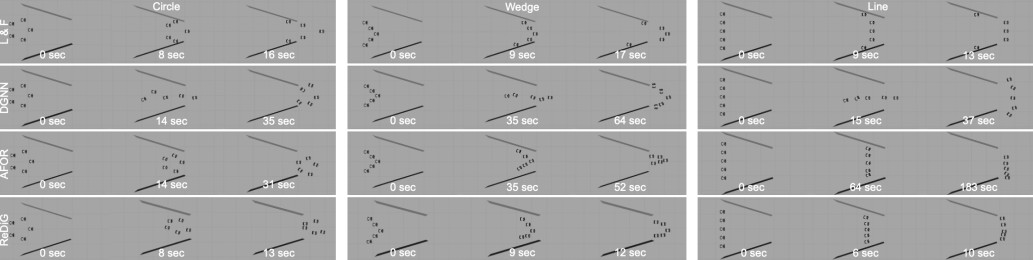

Figure 2: Qualitative results from Gazebo simulations on formation adaptation using Limo robots.

# 4 EXPERIMENTS

**Experimental Setups.** We comprehensively evaluate our ReDiG approach in three experimental settings: (1) a standard Gazebo simulation using ROS2, (2) a high-fidelity Unity-based 3D multi-robot simulator in ROS1, and (3) a physical robot team running ROS2. Each setup involves different differential-drive robot platforms (e.g., Limo and Warthog robots), and formation shapes (e.g., circle, wedge, and line). To follow the identical trajectories, we convert the linear velocity $\mathbf{a}_i$ into corresponding wheel velocities. All scenarios feature narrow corridors, the robot teams are required to navigate through confined spaces while dynamically adjusting their formation and preserving its overall structure. In simulation, robot positions and environmental obstacles are obtained directly from the Gazebo and Unity simulation. For physical experiments, each robot performs state estimation and environment mapping using a SLAM method (Zou et al. (2021)). See Appendix F.1 for method implementation and training details. Video demonstrations are available on our project website.

To demonstrate the effectiveness of ReDiG, we compare it with three prior methods for coordinated multi-robot navigation, including: (1) Leader and Follower method (**L&F**) Reily et al. (2020), where one robot is designated as the "leader" to guide the team, while the remaining robots act as "followers" that maintain the formation by tracking the leader's motion; (2) Decentralized GNN (**DGNN**) Blumenkamp et al. (2022), which employs an online RL framework to generate velocity commands for each robot, but does not account for formation control; and (3) Adaptive Formation with Oscillation Reduction (**AFOR**) Deng et al. (2025a), an online RL method that incorporates a spring-damper model to enable adaptive formation control, but does not account for trajectory smoothness and efficiency. See Appendix F.4 for details on baselines.

To quantitatively evaluate and compare ReDiG with other methods, we employ three metrics, including: (1) Success Rate (**SR**) measures the proportion of robots in the team that successfully reach their goal without collisions, (2) Travel Time (**TT**) represents the total navigation time used by the entire team to reach their goals. (3) Contextual Formation Integrity (**CFI**) measures the real-time adherence of the robots to their designated formation, based on a threshold $\delta$ that determines how strictly the formation shape must be preserved. See Appendix F.5 for details on CFI and its calculation.

**Results in Multi-Robot Simulations.** The qualitative results from the Gazebo simulation are presented in Figure 2. The L&F method, which relies on a predefined rigid formation, fails to navigate narrow corridors, as the outer robots collide with the walls. DGNN does not incorporate formation control, resulting in robots passing through the corridor sequentially without coordination. Both AFOR and our proposed ReDiG approach integrate a spring-damper model to enable formation adaptation. However, AFOR is built upon RL with step-wise action decisions, which result in jerky trajectories. In contrast, ReDiG generates significantly smoother trajectories by leveraging a diffusion-based policy. Since visualizations alone may not fully capture the impact of jerkiness, we further provide a quantitative analysis of motion trajectory smoothness in the discussion.

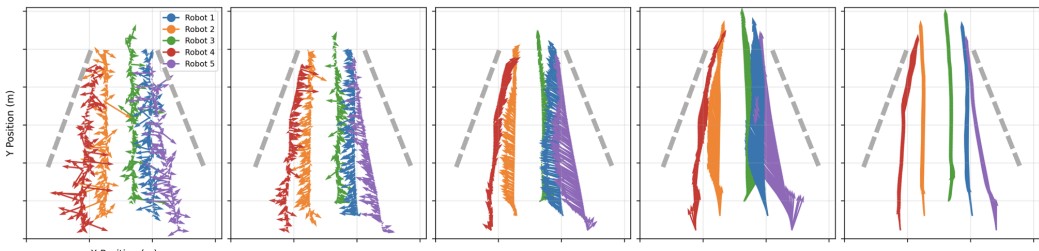

Figure 3: Visualization of ReDiG denoising for a wedge-formation team (left to right). Each colored arrow shows the velocity vector at a diffusion step of each robot. Dashed gray lines indicate obstacles.

We visualize the denoising process of our ReDiG approach for a robot team in wedge formation, as shown in Figure 3. Each arrow represents a velocity vector generated at different denoising steps. Starting from pure Gaussian noise, the action of each individual robot is progressively denoised into smooth, coordinated motion. This demonstrates ReDiG's ability to iteratively reconstruct meaningful actions that enable both smooth navigation and formation adaptation.

The quantitative results are shown in Table 1. The L&F method achieves a $60\%$ success rate due to its inability to adapt formations. DGNN, which lacks formation awareness, performs worst in the CFI metric. AFOR shows the longest travel time, especially in the line formation, due to the step-wise nature of multi-robot RL, which results in inefficient progress caused by jerky actions. Our proposed method addresses these limitations and achieves above $82\%$ CFI with the shortest travel time across all three formation shapes. ReDiG has slightly lower CFI scores compared to AFOR, which is a reasonable trade-off for the significant gain in efficiency. These results highlight the effectiveness of ReDiG in enabling smooth and efficient formation adaptation in complex environments.

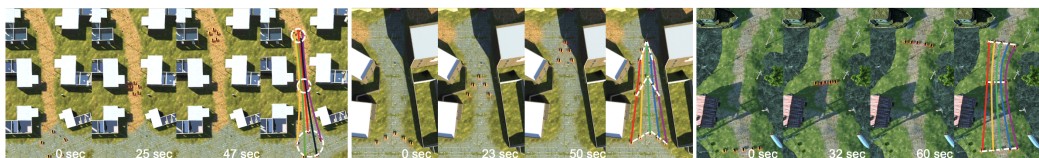

Figure 4: Qualitative results from Unity3D simulations using a team of differential-drive Warthog robots that follow circle, wedge, and line formations while navigating unstructured narrow corridors.

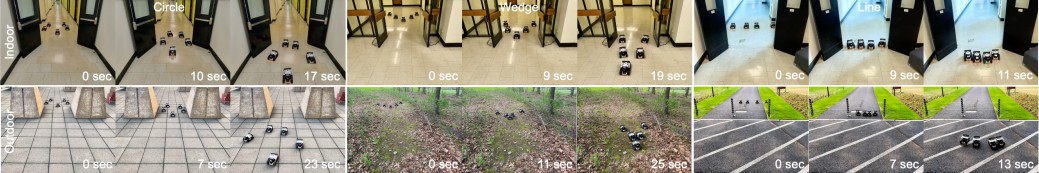

Figure 5: Qualitative results from real-world indoor and outdoor environments using varying numbers of Limo robots running ROS2 and communicating via Wi-Fi broadcasting.

Beyond Gazebo, we evaluate our approach in a Unity3D-based simulator integrated with ROS1 for multi-robot perception and control. These outdoor environments include extended narrow pathways between buildings and uneven flooded terrain, requiring long, curved trajectories with adaptive formation control. As shown in Figure 4, ReDiG enables Warthog teams to adapt their formation to environmental constraints, navigating smoothly and reaching goals without collisions. In circle formation scenarios, the team traverses narrow corridors by continuously reshaping the formation for tighter spaces. These results highlight ReDiG's effectiveness in achieving smooth, adaptive formation control in complex environments. See Appendix G for additional qualitative results.

**Validation on Physical Robot Teams.** We validate ReDiG through case studies using physical differential-drive Limo robots, each equipped with an onboard Intel NCU i7 processor for real-time execution. The robots run ROS2 and coordinate via Wi-Fi-based broadcasting for decentralized communication. Experiments were conducted in both indoor and outdoor environments. As shown in Figure 5, our method enables teams of varying sizes to smoothly adapt formation while navigating

narrow indoor spaces. The outdoor experiments include narrow passages bordered by bollards, scattered trees, and roadblocks. Notably, in wedge formation scenarios, ReDiG effectively guided the team through uneven forest terrain where wheel slippage introduced movement uncertainty. The results further demonstrate the effectiveness of our approach in enabling coordinated navigation with formation adaptation, as well as its adaptability to unfamiliar environments.

# 5 DISCUSSION

**Importance of Inter-Robot Message Passing for Coordination.** To understand how graph learning facilitates team coordination, we conduct a message importance analysis. The graph network uses learnable weight $\mathbf{W}^h$ to compute team embeddings $\mathbf{h}_i$ through message passing. We quantify the importance of messages exchanged between robots by calculating the Euclidean norm of the message vectors $||\mathbf{W}^h(\mathbf{z}_j - \mathbf{z}_i)||_2$. Figure 6 visualizes message importance between robot pairs during training. Early in training, robot 4 distributes attention uniformly

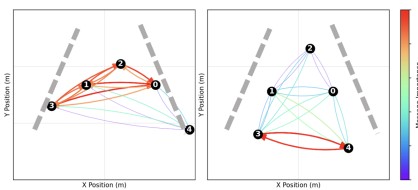

Figure 6: Comparison of inter-robot message importance over training.

across teammates, lacking awareness of team structure, which eventually leads to a collision with the wall. By the end of training, robots 3 and 4, as the outermost in the formation and key to controlling team size, assign the highest importance to each other's messages. This verifies the effectiveness of graph learning in enabling team coordination and formation adaptation.

**Ablation Study on Motion Trajectory Smoothness.** We conduct an ablation study to evaluate the role of the diffusion and the spring–damper model in enhancing motion trajectory smoothness. We use the jerk metric Gasparetto & Zanotto (2007) to evaluate smoothness, which measures the

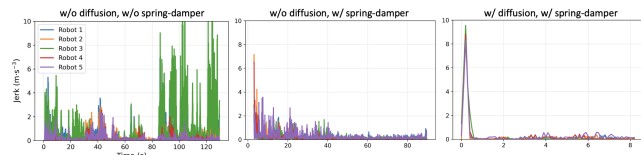

Figure 7: Ablation to diffusion and spring-damper model.

rate of change of acceleration over time. For each robot $i$, the jerk is defined as $\mathbf{j}_i(t) = \frac{d^3\mathbf{p}_i(t)}{dt^3}$. Figure 7 shows the jerk profiles over time for a team of five robots. The first subfigure shows the case without diffusion and without the spring-damper model, where the jerk is the highest. The second corresponds to using the spring-damper model without diffusion, the damping effect reduces jerk, but noticeable fluctuations remain. The third combines both diffusion and the spring-damper model, where iterative refinement of robot actions achieves the lowest jerk. These results highlight the effectiveness of the diffusion model in producing smooth individual robot trajectories.

**Scalable Time Complexity.** ReDiG is computationally efficient during training and supports real-time performance in decentralized execution. The time complexity is $O(n^2)$ for training and $O(n+T_d)$ for execution, where $n$ is the number of robots and $T_d$ is the number of diffusion steps. See Appendix E for details. The analytical time complexity indicates that our approach naturally generalizes to larger teams and remains scalable as the number of robots increases. Training remains efficient because computations are parallelized across batches, while during decentralized execution, each robot only performs local computations from a bounded set of neighbors, which ensures scalable execution. In practice, a team of 5 robots achieves an execution runtime of 7.81 ms per step in simulation (128 Hz), and 66.7 ms per step in physical multi-robot systems (15 Hz). See Appendix F.7 for details.

# 6 CONCLUSION

In this paper, we propose ReDiG to enable decentralized coordinated multi-robot navigation with smooth formation adaptation. ReDiG is built upon a unified learning paradigm, including graph learning for decentralized coordination to enable formation adaptation, diffusion models for generating smooth trajectories for individual robots, and online reinforcement learning to refine noisy demonstrations, which enables robot synchronization and guides effective diffusion training. Results from extensive experiments show that ReDiG enables smooth formation adaptation and achieves state-of-the-art performance in coordinated multi-robot navigation.

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

## A    AUTHOR DECLARATIONS

**Ethics Statement.**    We confirm that this research complies with the ICLR Code of Ethics, available at https://iclr.cc/public/CodeOfEthics.

**Reproducibility Statement.**    We confirm that this paper discloses all information necessary to reproduce the main experimental results that support the claims and conclusions. Implementation details and hyperparameters are provided in Appendix F.1, and Section 3.2.3 describes the training and execution process in detail.

**Declaration of LLM Usage.**    We confirm that large language models (LLMs) were used only to assist with writing. They did not influence the methodology, scientific rigor, or originality of this research.

## B    LIMITATIONS

ReDiG has several limitations that open up promising directions for future research.

First, the current approach requires retraining when switching between different formation types (e.g., wedge, line, circle). While this allows for formation-specific optimization, it limits the flexibility of the system when dynamic formation transitions are needed. In future work, we plan to incorporate formation-type conditioning into the policy and design an additional team-level decision-making module that selects the most suitable formation based on the current environment. This would allow the model to generalize across multiple formations without retraining.

Second, ReDiG does not rely on team-level expert demonstrations (e.g., adaptive formation control). Instead of generating random trajectories for individual robots, we initialize training with trajectories derived from path-planning algorithms such as A* or RRT. This strategy facilitates faster convergence, but it also introduces a dependency on predefined single-robot planning methods. We plan to reduce this requirement by exploring self-supervised or curriculum learning strategies that can learn effective behavior from scratch through environment interaction.

Third, ReDiG's current implementation is evaluated with a team of ground robots operating on a 2D surface. While this provides a controlled and reproducible testbed for ground vehicles, it does not capture the full complexity of real-world scenarios involving uneven terrain, varying elevation, or multi-layered environments, nor does it account for other types of robots operating in full 3D spaces. As a next step, we plan to extend our approach to 3D environments and teams of mixed ground and aerial robots by incorporating terrain-aware and altitude-adaptive control to better handle unstructured outdoor settings.

## C    PROOF OF THEOREMS

We present a diffusion-based online reinforcement learning paradigm to improve the diffusion models. The diffusion model serves as the actor network, generating actions that are stored in a replay buffer. A deep Q-network functions as the critic, evaluating these actions and providing value estimates. The critic uses gradient ascent to refine the diffusion-generated actions toward higher Q-values, and these improved actions are then used as training targets for the diffusion model. This creates an iterative improvement cycle where the diffusion model learns to generate increasingly better actions.

We present upper bounds for both components, including a diffusion model convergence guarantee that ensures reliable action generation given refined training targets, while the critic convergence bound ensures accurate value estimation of actions. With these convergence guarantees established, we can apply gradient ascent to refine these actions.

We summarize the mathematical notations used throughout the paper in Table 2. These notations will also be used to show mathematical proofs here.

Table 2: Mathematical notations (in order of appearance within the main paper)

| Variable | Definition |
|---|---|
| $\mathcal{V}$ | Node set |
| $\mathbf{E}$ | Edge matrix |
| $\mathbf{p}_i$ | Position |
| $\mathbf{q}_i$ | Velocity |
| $\mathbf{g}_i$ | Goal |
| $d_i$ | Obstacle proximity |
| $\mathbf{s}_i$ | State |
| $\mathbf{a}_i$ | Action |
| $\mathbf{z}_i$ | Individual state embedding |
| $\mathbf{W}^z$ | Learnable weight matrix for individual state embedding |
| $\mathbf{h}_i$ | Team state embedding |
| $\mathbf{W}^h$ | Learnable weight matrix for team state embedding |
| $\mathbf{a}_i^0$ | Ground-truth action |
| $k$ | Diffusion steps |
| $\beta^k$ | Variance of noise |
| $\alpha^k$ | Retention factor |
| $\sigma^k$ | Noise scale |
| $\lambda^k$ | Weighting coefficient |
| $d_{i,j}$ | Expected distance between $i$-th and $j$-th robot |
| $p_{i,j}$ | Actual distance between $i$-th and $j$-th robot |
| $q_{i,j}$ | Relative velocities between $i$-th and $j$-th robot |
| $R$ | Reward |
| $\omega$ | Hyperparameter for spring-damper model |
| $\mathbf{y}_i$ | Target value |
| $\gamma$ | Discount factor |
| $\eta$ | Step size for action refinement |
| $\mathcal{M}$ | Replay buffer |
| $\delta$ | Threshold for CFI metric |
| **Function** | **Definition** |
| $p(\mathbf{a}_i|\mathbf{h}_i)$ | Reverse denoising distribution conditioned on graph embedding |
| $q(\mathbf{a}_i^k|\mathbf{a}_i^{k-1})$ | Forward noising distribution |
| $\epsilon_\theta(\mathbf{a}_i^k, k)$ | Diffusion function |
| $\hat{p}_\theta(\mathbf{a}_i^0|\mathbf{h}_i)$ | Learned action distribution |
| $p(\mathbf{a}_i^0|\mathbf{h}_i)$ | True action distribution |
| $q(\mathbf{a}_i^K)$ | Marginal distribution |
| $Q(\mathbf{h}_i, \mathbf{a}_i)$ | Deep Q function |

## C.1 PROOF OF THEOREM 1

We first present the proof of the upper bound for the diffusion model on graphs for coordinated multi-robot navigation. Our objective is to bound the loss defined in Eq.(1), which quantifies how well the learned conditional distribution $\hat{p}_\theta(\mathbf{a}_i^0|\mathbf{h}_i)$ approximates the true distribution $p(\mathbf{a}_i^0|\mathbf{h}_i)$, where $\mathbf{a}_i^0$ denotes the clean action for robot $i$, and $\mathbf{h}_i$ is the graph embedding that encodes the team context.

The forward process progressively adds Gaussian noise to the clean action $\mathbf{a}_i^0$ over $K$ steps, which is defined as:

$$q(\mathbf{a}_i^k|\mathbf{a}_i^{k-1}) = \mathcal{N}(\mathbf{a}_i^k; \sqrt{1-\beta^k}\mathbf{a}_i^{k-1}, \beta^k\mathbf{I}),$$

where $\{\beta^k\}_{k=1}^K$ is a noise variance schedule with $\beta^k \in (0,1)$. By composing the forward steps, we obtain the marginal distribution, which is defined as:

$$q(\mathbf{a}_i^k|\mathbf{a}_i^0) = \mathcal{N}(\mathbf{a}_i^k; \sqrt{\alpha^k}\mathbf{a}_i^0, (1-\alpha^k)\mathbf{I}),$$

where $\alpha^k = \prod_{j=1}^k (1-\beta^j)$ is the cumulative noise coefficient.

The reverse process reconstructs clean actions from noisy inputs using a neural network $\epsilon_\theta(\mathbf{a}_i^k, k|\mathbf{h}_i)$ that predicts the noise added at step $k$, which is defined as:

$$\mathbf{a}_i^{k-1} = \lambda^k \left( \mathbf{a}_i^k - \alpha^k \epsilon_\theta(\mathbf{a}_i^k, k|\mathbf{h}_i) \right) + \sigma^k z, \tag{3}$$

where $z \sim \mathcal{N}(0, \mathbf{I})$, and $\lambda^k = 1/\sqrt{1 - \beta^k}$ and $\sigma^k$ are derived from $\beta^k$. This process defines an implicit distribution $\hat{p}_\theta(\mathbf{a}_i^0|\mathbf{h}_i)$ by sampling $\mathbf{a}_i^K \sim \mathcal{N}(0, \mathbf{I})$ and applying $K$ reverse steps to generate $\mathbf{a}_i^0$.

We treat this diffusion process as a latent-variable model with latent variables $\mathbf{a}_i^{1:K}$. Using the forward process $q(\mathbf{a}_i^{1:K}|\mathbf{a}_i^0)$ as the variational posterior, we apply the evidence lower bound (ELBO) from variational inference, which is defined as:

$$\log p(\mathbf{a}_i^0|\mathbf{h}_i) \geq \mathbb{E}_q \left[ \log p_\theta(\mathbf{a}_i^{0:K}|\mathbf{h}_i) - \log q(\mathbf{a}_i^{1:K}|\mathbf{a}_i^0) \right]. \tag{4}$$

This leads to an upper bound on the KL divergence between the learned and true conditional distributions, which is defined as:

$$D_{\mathrm{KL}}(\hat{p}_\theta(\mathbf{a}_i^0|\mathbf{h}_i) \,\|\, p(\mathbf{a}_i^0|\mathbf{h}_i)) \leq -\mathbb{E}_q \left[ \log p_\theta(\mathbf{a}_i^{0:K}|\mathbf{h}_i) - \log q(\mathbf{a}_i^{1:K}|\mathbf{a}_i^0) \right]. \tag{5}$$

**Prior Mismatch.** The ELBO can be decomposed into a sum of KL divergences between the reverse model $p_\theta(\mathbf{a}_i^{k-1}|\mathbf{a}_i^k, \mathbf{h}_i)$ and the true reverse posterior $q(\mathbf{a}_i^{k-1}|\mathbf{a}_i^k, \mathbf{a}_i^0)$, which is defined as:

$$\sum_{k=1}^K \mathbb{E}_{q(\mathbf{a}_i^k|\mathbf{a}_i^0)} \left[ D_{\mathrm{KL}} \left( q(\mathbf{a}_i^{k-1}|\mathbf{a}_i^k, \mathbf{a}_i^0) \,\|\, p_\theta(\mathbf{a}_i^{k-1}|\mathbf{a}_i^k, \mathbf{h}_i) \right) \right] + D_{\mathrm{KL}} \left( q(\mathbf{a}_i^K) \,\|\, p(\mathbf{a}_i^K) \right). \tag{6}$$

The last term represents the KL divergence between the marginal of the final noisy state and the Gaussian prior. When $q(\mathbf{a}_i^K)$ poorly matches the prior $p(\mathbf{a}_i^K) = \mathcal{N}(0, \mathbf{I})$, the initial samples used in the reverse process may be unrealistic, resulting in infeasible actions.

**Matching Error.** The true reverse posterior $q(\mathbf{a}_i^{k-1}|\mathbf{a}_i^k, \mathbf{a}_i^0)$ is Gaussian with mean:

$$\tilde{\mu}_i^k = \frac{\sqrt{\alpha^{k-1}}\beta^k}{1 - \alpha^k} \mathbf{a}_i^0 + \frac{\sqrt{1 - \beta^k}(1 - \alpha^{k-1})}{1 - \alpha^k} \mathbf{a}_i^k. \tag{7}$$

The model distribution $p_\theta(\mathbf{a}_i^{k-1}|\mathbf{a}_i^k, \mathbf{h}_i)$ is also Gaussian, with mean:

$$\mu_\theta(\mathbf{a}_i^k, k|\mathbf{h}_i) = \lambda^k \left( \mathbf{a}_i^k - \alpha^k \epsilon_\theta(\mathbf{a}_i^k, k|\mathbf{h}_i) \right). \tag{8}$$

We express the noisy input $\mathbf{a}_i^k$ using its forward reparameterization:

$$\mathbf{a}_i^k = \sqrt{\alpha^k}\mathbf{a}_i^0 + \sqrt{1 - \alpha^k}\,\epsilon, \tag{9}$$

where $\epsilon \sim \mathcal{N}(0, \mathbf{I})$ is the true noise. Substituting this into both means, we find that aligning the coefficients of $\epsilon$ leads to the condition:

$$\epsilon_\theta(\mathbf{a}_i^k, k|\mathbf{h}_i) \approx \epsilon. \tag{10}$$

Hence, minimizing the squared error between the predicted and true noise:

$$\mathbb{E}_{\mathbf{a}_i^0, \epsilon, k} \left[ \left\| \epsilon - \epsilon_\theta(\mathbf{a}_i^k, k|\mathbf{h}_i) \right\|_2^2 \right], \tag{11}$$

is equivalent to minimizing the KL divergence between the reverse distributions at each step. This ensures that the model-predicted mean $\mu_\theta(\mathbf{a}_i^k, k|\mathbf{h}_i)$ closely approximates the true mean $\tilde{\mu}_i^k$, producing accurate and smooth action reconstructions.

**Discretization Error.** The reverse process can also be interpreted as a discretization of a continuous-time reverse stochastic differential equation (SDE). While the theoretical formulation assumes an infinite number of infinitesimal denoising steps, practical implementations must approximate this process using a finite number of discrete transitions. This numerical approximation introduces discretization error. Under standard regularity assumptions, such as Lipschitz continuity of the score function and bounded second moments, the discretization error is bounded, which is defined as:

$$\mathcal{O}\left( \sum_{k=1}^K (\beta^k)^2 \right). \tag{12}$$

In multi-robot navigation, this error may lead to deviations from smooth or coordinated behaviors present in expert demonstrations, particularly when using a small number of steps or a poorly chosen noise schedule.

Combining all components, we obtain the following upper bound of the loss defined in Eq.(1) between the learned and true conditional distributions:

$$
D_{\mathrm{KL}}\big(\hat{p}_\theta(\mathbf{a}_i^0|\mathbf{h}_i)\,\|\,p(\mathbf{a}_i^0|\mathbf{h}_i)\big) \leq \overbrace{D_{\mathrm{KL}}\big(q(\mathbf{a}_i^K)\,\|\,p(\mathbf{a}_i^K)\big)}^{\text{Prior mismatch}}
$$

$$
+ \overbrace{\sum_{k=1}^{K} \mathbb{E}_{\mathbf{a}_i^0,\epsilon,k}\left[\left\|\epsilon - \epsilon_\theta(\mathbf{a}_i^k, k|\mathbf{h}_i)\right\|_2^2\right]}^{\text{Matching error}} + \overbrace{\mathcal{O}\left(\sum_{k=1}^{K}(\beta^k)^2\right)}^{\text{Discretization error}} \quad (13)
$$

This analysis supports the design of our diffusion model and helps identify which source of error may be affecting trajectory quality. For example, if the prior mismatch is large, the initial sampling distribution may need to be improved. If the matching error is high, the model architecture or training process may require adjustment. If the discretization error is significant, increasing the number of diffusion steps may help.

## C.2 Proof of Theorem 2

We then present the proof in Theorem 2 for the upper bound of the critic loss defined in Eq.(2).

Let $\gamma \in (0,1)$ and assume $|R(s_i, a_i)| \leq R_{\max}$ for all state-action pairs. For any bounded $Q$, we define the Bellman evaluation operator $T$ as follows:

$$
(TQ)(\mathbf{h}_i, \mathbf{a}_i) \;=\; R(s_i, a_i) + \gamma\,\mathbb{E}_{(\mathbf{h}_i', \mathbf{a}_i')\sim\mu}\Big[Q(\mathbf{h}_i', \mathbf{a}_i')\Big]
$$

where $\mu$ represents the data distribution in the replay buffer, and its unique fixed point $Q^*$ satisfying $Q^* = TQ^*$. We further define the linear operator as follows:

$$
(Pf)(\mathbf{h}_i, \mathbf{a}_i) \;:=\; \mathbb{E}_{(\mathbf{h}_i', \mathbf{a}_i')\sim\mu|\mathbf{h}_i, \mathbf{a}_i}\big[f(\mathbf{h}_i', \mathbf{a}_i')\big] \quad (14)
$$

where $f$ is any bounded function, $P$ is the one-step transition operator under the replay distribution $\mu$, and $Pf$ denotes the function obtained by taking the conditional expectation of $f$ at the next step.

Let $\{\hat{Q}_m\}_{m=0}^{M}$ be the sequence of the estimates from critics after $M$ training iterations, where $\hat{Q}_m$ is the value function produced after the $m$-th training iteration. We define the one-step Bellman regression residual $e_m$ at iteration $m$ as follows:

$$
e_m \;:=\; \hat{Q}_m - T\hat{Q}_{m-1} \quad (15)
$$

Then, the maximum residual across the first $M$ iteration is defined as:

$$
S \;:=\; \max_{1\leq m\leq M} \|e_m\| \quad (16)
$$

where $\|\cdot\|$ denotes the $L_1$ norm under the replay buffer distribution.

We assume the norm $\|\cdot\|$ is compatible with one-step propagation, meaning that applying the transition operator $P$ to any bounded function does not increase its norm by more than a fixed constant $C$, that is,

$$
\|Pf\| \;\leq\; C\,\|f\|, \quad (17)
$$

Equivalently, the operator norm of $P$ (as a linear map on bounded functions) satisfies $\|P\|_{\mathrm{op}} \leq C$. We allow the transition operator to vary with iteration and write $P^{(k)}$ for the one-step operator used at iteration $k$.

For iteration index $m \geq 1$ and $0 \leq j \leq m-1$, we define the family of $j$-step composition operators as:

$$
\mathcal{P}_{m,j} \;:=\; \left\{ P^{(m-1)}P^{(m-2)}\cdots P^{(m-j)} \right\}
$$

We denote by $\|\cdot\|_{\mathrm{op}}$ the operator norm induced by $\|\cdot\|$.

Since the one-step transition operator is non-expansive in the chosen norm (e.g., $\|P\|_{\mathrm{op}} \leq 1$ under the sup-norm), multi-step propagations remain bounded and the discounted series converges. We therefore assume that the cumulative effect of multi-step propagation is bounded as follows:

$$(1 - \gamma)^2 \sum_{m=1}^{\infty} \gamma^{m-1} \sum_{j=0}^{m-1} \sup_{A \in \mathcal{P}_{m,j}} \|A\|_{\mathrm{op}} \leq C \tag{18}$$

where each $A \in \mathcal{P}_{m,j}$ is a $j$-step composition of one-step transition operators from iterations $m - j$ to $m - 1$, and sup denotes the supremum (least upper bound), i.e., the maximum operator norm over all such $A$.

Let $\Delta_m := \hat{Q}_m - Q^*$. Since $Q^* = TQ^*$ and $TQ = R + \gamma PQ$, we have the one-step bound:

$$|\Delta_m| = \left|e_m + (T\hat{Q}_{m-1} - TQ^*)\right| = \left|e_m + \gamma P(\hat{Q}_{m-1} - Q^*)\right|$$
$$\leq |e_m| + \gamma\, P\, |\Delta_{m-1}| \tag{19}$$

where the inequality uses positivity of $P$ (if $f \geq 0$ then $Pf \geq 0$).

Unrolling equation 19 for any $M \geq 1$, we have:

$$|\Delta_M| \leq \sum_{m=1}^{M} \gamma^{m-1} P^{m-1} |e_{M-m+1}| + \gamma^M P^M |\Delta_0| \tag{20}$$

where $P^j$ denotes the $j$-fold composition of the one-step transition operator.

After applying the norm $\|\cdot\|$ to the path expansion in equation 20 and using the one-step coverage assumption, we obtain:

$$\|\Delta_M\| \leq \sum_{m=1}^{M} \gamma^{m-1} \|P^{m-1}|e_{M-m+1}|\| + \gamma^M \|P^M |\Delta_0|\| \tag{21}$$

To further bound these terms, we note that repeated applications of $P$ can be represented by the operator family $\mathcal{P}_{m,j}$. Using this notation, we have:

$$\|\Delta_M\| \leq 2 \sum_{m=1}^{M} \gamma^{m-1} \left( \sum_{j=0}^{m-1} \sup_{A \in \mathcal{P}_{m,j}} \|A\|_{\mathrm{op}} \right) S + 2\gamma^M \left( \sum_{j=0}^{M-1} \sup_{A \in \mathcal{P}_{M,j}} \|A\|_{\mathrm{op}} \right) \|\Delta_0\| \tag{22}$$

where $S = \max_{1 \leq m \leq M} \|e_m\|$ is the largest one-step residual.

**Statistical Error.** By the multi-step propagation bound in Eq. (18), we have:

$$\sum_{m=1}^{\infty} \gamma^{m-1} \left( \sum_{j=0}^{m-1} \sup_{A \in \mathcal{P}_{m,j}} \|A\|_{\mathrm{op}} \right) \leq \frac{C}{(1 - \gamma)^2} \tag{23}$$

Since the finite sum up to $M$ is bounded by the infinite sum, and each residual $\|e_m\|$ is at most $S$ by definition, while the factor 2 comes from the pairing step in Eq. (22), multiplying by $2S$ yields:

$$2 \sum_{m=1}^{M} \gamma^{m-1} \left( \sum_{j=0}^{m-1} \sup_{A \in \mathcal{P}_{m,j}} \|A\|_{\mathrm{op}} \right) S \leq \frac{2\,\gamma\,C}{(1 - \gamma)^2} S \tag{24}$$

where the factor $\gamma$ is absorbed into the definition of the constant $C$.

**Algorithmic Error.** Since the true value function $Q*$ satisfies the Bellman equation, and rewards are bounded by $|R(s_i, a_i)| \leq R_{\max}$, the total return is bounded by the geometric series as follows:

$$|Q^*(\mathbf{h}_i, \mathbf{a}_i)| \leq \sum_{t=0}^{\infty} \gamma^t R_{\max} = \frac{R_{\max}}{1 - \gamma}$$

Hence, for any norm consistent with pointwise bounds (e.g., $L_1$ or $L_\infty$), we have:

$$\|Q^*\| \leq \frac{R_{\max}}{1-\gamma} \tag{25}$$

Therefore, the initialization gap is bounded as:

$$\|\Delta_0\| \leq \|\hat{Q}_0\| + \|Q^*\| \leq \frac{2R_{\max}}{1-\gamma} \tag{26}$$

For the finite-iteration error, using bounded operator norms and a conservative geometric bound, we obtain:

$$2\gamma^M \left( \sum_{j=0}^{M-1} \sup_{A \in \mathcal{P}_{M,j}} \|A\|_{\mathrm{op}} \right) \|\Delta_0\| \leq \frac{4\gamma^{M+1}}{(1-\gamma)^2} R_{\max} \tag{27}$$

This term decays exponentially as $M \to \infty$, ensuring convergence.

By combining both statistical error in Eq. (24) and algorithmic error in Eq. (27), we obtain the complete error bound, which is defined as follows:

$$\|\hat{Q}_M - Q^*\| \leq \underbrace{\frac{2\gamma C}{(1-\gamma)^2} S}_{\text{statistical error}} + \underbrace{\frac{4\gamma^{M+1}}{(1-\gamma)^2} R_{\max}}_{\text{algorithmic error}} \tag{28}$$

---

**Algorithm 1:** Unified Training for Reinforced Diffusion on Graph (ReDiG)

---

**Input** : Team graph representation $\mathcal{G}$, individual robot state $\mathbf{s}$, replay buffer $\mathcal{M}$
**Output**: Trained graph neural network $\phi$, diffusion model $\psi$, and critic networks $Q_1, Q_2$

1   Initialize all learnable components in ReDiG, including $\mathbf{W}^z$, $\mathbf{W}^h$, $\epsilon_\theta$, $Q_1, Q_2$;
2   Generate individual ground-truth actions using path-planning policies, and store $(\mathbf{s}_i, \mathbf{a}_i, R_i, \mathbf{s}'_i)$ in $\mathcal{M}$;
3   **while** *Not converged* **do**
4      Use the decentralized graph neural network to generate team embeddings $\mathbf{h}_i = \phi(\mathcal{G})$;
5      Use diffusion to generate actions $\{\mathbf{a}_i\}^n = \{\psi(\mathbf{h}_i)\}^n$;
6      Compute the gradient of diffusion loss $\nabla_{\mathbf{a}_i^k} \epsilon_\theta(\mathbf{a}_i^k, k|\mathbf{h}_i)$ according to Eq. (1);
7      Update denoising network $\epsilon_\theta$ in the diffusion model $\psi$ according to the gradient $\nabla_{\mathbf{a}_i^k} \epsilon_\theta(\mathbf{a}_i^k, k|\mathbf{h}_i)$;
8      Apply $\{\mathbf{a}_i\}^n$ in the environment and store the transition $(\mathbf{s}_i, \mathbf{a}_i, R_i, \mathbf{s}'_i)$ to $\mathcal{M}$;
9      Sample batch $(\mathbf{s}_i, \mathbf{a}_i, R_i, \mathbf{s}'_i)$ from $\mathcal{M}$;
10     Compute the gradient of the critic loss $\nabla_{\mathbf{a}_i} Q(\mathbf{h}_i, \mathbf{a}_i)$ according to Eq. (2);
11     Refine the action $\mathbf{a}_i = \mathbf{a}_i + \eta \nabla_{\mathbf{a}_i} Q(\mathbf{h}_i, \mathbf{a}_i)$;
12     Update GNN $\mathbf{W}^z$ and $\mathbf{W}^h$ according to the gradient $\nabla_{\mathbf{a}_i^k} \epsilon_\theta(\mathbf{a}_i^k, k|\mathbf{h}_i)$ and $\nabla_{\mathbf{a}_i} Q(\mathbf{h}_i, \mathbf{a}_i)$;
13     Update $Q_1$ and $Q_2$ according to the gradient $\nabla_{\mathbf{a}_i} Q(\mathbf{h}_i, \mathbf{a}_i)$
14   **end**
15   **return** $\mathbf{W}^z$, $\mathbf{W}^h$, $\epsilon_\theta$, $Q_1, Q_2$

---

# D   Unified Training Algorithm for ReDiG

Formally, ReDiG full training algorithm is presented in Algorithm 1. In line 2, we use an individual search-based path planning algorithm (e.g., $A^*$, RRT) to generate individual robot trajectories that focus solely on goal-reaching, without accounting for coordinated navigation or synchronized formation adaptation. These trajectories serve as initial ground-truth supervision for training the diffusion model and are stored in the replay buffer $\mathcal{M}$. In line 4, we compute the team embedding given the decentralized graph neural network $\phi$ with the input of the team graph representation. Given the team embedding, we compute the actions of individual robots according to the diffusion model $\psi$ in line 5. In lines 6-7, the denoising network $\epsilon_\theta$ in the diffusion model $\psi$ is updated by minimizing the diffusion loss defined in Eq. (1) given the computed gradient. In line 8, we collect all actions $\{\mathbf{a}_i\}^n$ generated by all the robots and apply the actions in the environments, and the resulting transitions are stored in the replay buffer $\mathcal{M}$. Then, we uniformly sample a batch from the replay buffer in line 9 and compute the gradient of the critic loss according to Eq. (2) in line 10. In lines 11-13, we refine the actions, update the graph neural network, and critic networks given the computed gradients.

# E   TIME COMPLEXITY ANALYSIS

## E.1   TIME COMPLEXITY ANALYSIS AND SCALABILITY OF ReDiG

*Training time complexity* is dominated by $O(n^2)$, where $n$ is the number of robots. The decentralized graph network generates state embeddings has an $O(L_g n^2)$ complexity, where $L_g$ is the number of GNN layers, $n^2$ is the number of edges computed for communications. The denoising network training has an $O(T_d B_d L_d)$ complexity, where $T_d$ is the number of denoising steps, $B_d$ is the sampled batch size, and $L_d$ is the number of denoising network layers. The critic network training has an $O(B_c L_c n)$ complexity, where $B_c$ is the sampled batch size, $L_c$ is the number of critic network layers. The action refinement using gradient ascent has an $O(G_a B_a n)$ complexity, where $G_a$ is the number of gradient steps, $B_a$ is the sampled batch size, and each gradient step requires critic evaluation with $O(n)$ complexity. Combining all terms, the overall complexity for training is $O(I(n^2 + T_d B_d L_d + B_c L_c n + G_a B_a n))$, where $I$ is the number of training iterations. *Execution time complexity* is dominated by $O(n)$. The complexities of embeddings generation from graph and actions generation from diffusion policy are $O(L_g n)$ and $O(T_d L_d)$, the overall execution time complexity is $O(n + T_d)$.

The above time complexity provides a direct analytical measure of the scalability of our approach. Training involves quadratic interactions $O(n^2)$ due to pairwise message passing in the graph encoder, but this cost is amortized through batch parallelization, making it tractable even for large teams. During decentralized execution, the complexity reduces to $O(n + T_d)$, where the linear term reflects local neighbor aggregation and the constant term corresponds to the diffusion process. This separation of costs highlights that the per-robot computation remains bounded, and with decentralized parallelization, the effective execution complexity is dominated by the diffusion steps rather than the team size. Thus, the complexity analysis demonstrates that ReDiG is scalable and readily generalizes to larger robot teams.

## E.2   TIME COMPLEXITY ANALYSIS OF BASELINE METHODS

We also analyze the time complexity of baseline methods for comparison. While ReDiG exhibits a similar training complexity to these baselines, it achieves a lower execution complexity, enabling more efficient and scalable real-time performance.

**Leader-Follower.**   *Time complexity* is dominated by $O(n^2)$, where $n$ is the number of robots. The leader's path planning has a complexity of $O(P)$, where $P$ depends on the path planning algorithm used (e.g., A* or RRT). For A*, the complexity is typically $O(b^d)$, where $b$ is the branching factor and $d$ is the depth of the optimal solution. For RRT, it is typically $O(k \log k)$, where $k$ is the number of sampled nodes. Each follower computes its position relative to the leader and potentially other followers to maintain formation, resulting in $O(n-1)$ complexity per follower. For $n-1$ followers, the total coordination complexity is $O((n-1)^2) = O(n^2)$. Therefore, the overall time complexity is $O(P + n^2)$.

**Online Reinforcement Learning with PPO.**   *Training time complexity* is dominated by $O(n^2)$, where $n$ is the number of robots. The GNN has an $O(L_m T_p n^2)$ complexity, where $L_m$ is the number of layers in the GNN and $T_p$ is the number of training iterations using Proximal Policy Optimization (PPO). The control network training has an $O(T_p(Bn^2 + IBn))$ complexity, where $B$ is the number of PPO rollouts to interact with the environment in each iteration, and $I$ is the number of PPO training epochs. $O(Bn^2)$ accounts for computing the advantage function, and $O(IBn)$ for updating the policy. Combining all terms, the overall complexity for training is $O(HL_h T_h Dn^2 + L_m T_p n^2 + T_p(Bn^2 + IBn))$. *Execution time complexity* is dominated by $O(n^2)$. The complexities of the GNN and control networks are $O(L_m n^2)$ and $O(n)$, respectively.

# F  EXPERIMENTAL DETAILS

## F.1  ADDITIONAL IMPLEMENTATION DETAILS FOR REDiG

To implement ReDiG, we construct the robot team graph by connecting neighboring robots that are within a 2.0-meter spatial radius. The graph learning consists of an encoder with a weight matrix $\mathbf{W}^z$ of dimension $6 \times 64$, followed by a single GNN layer with a weight matrix $\mathbf{W}^h$ of dimension $64 \times 64$. The diffusion model employs a denoising network $\epsilon_\theta$ set to a dimension of $40 \times 256$ with Mish activation functions. The critic includes two separate but identical value networks, $Q_1$ and $Q_2$, both set to the dimension of $8 \times 256$. In the spring-damper model of ReDiG, the hyper-parameter $\omega = 0.5$ is used to balance the contributions of the spring and damper forces. We generate synthetic data to train our ReDiG approach by randomly generating robot positions within a given team formation. We employ the Adam optimizer Kingma (2014) to train all three learning components, including the graph neural network, denoising network, and critic networks. For initial exploration, we use the individual path planning algorithm to generate individual robot trajectories, which results in a buffer of 5,000 trajectory instances.

## F.2  HYPERPARAMETERS

Training is conducted on a machine equipped with a 16-core Intel i9 CPU, 32GB of RAM, and an NVIDIA RTX 4090 GPU. The entire training process, involving graph networks, diffusion denoising network, and critic value networks, is trained over 30,000 epochs. We further provide the hyperparameters used for ReDiG in our experiments, as shown in Table 3.

Table 3: ReDiG Hyperparameters

| Hyperparameter | Value | Hyperparameter | Value |
|---|---|---|---|
| Hidden layers | 3 | Hidden units per layer | 256 |
| Policy/Value network activation | Mish | Batch size | 256 |
| Discount factor | 0.99 | Target smoothing | 0.005 |
| Actor learning rate | $3 \times 10^{-4}$ | Critic learning rate | $3 \times 10^{-4}$ |
| GNN learning rate | $3 \times 10^{-4}$ | Actor–critic grad norm | 2.0 |
| Replay buffer size | $1 \times 10^6$ | Reward scaling factor | 0.01 |
| Diffusion steps | 100 | Beta schedule | Cosine |
| Noise ratio | 1.0 | GNN message dimension | 32 |
| GNN communication range | 2.0 | GNN activation | ReLU |

## F.3  REWARD DESIGN

We present the reward design of the spring-damper model used in ReDiG to enable adaptive formation control as follows:

$$R^{adp} = \sum_{\mathbf{v}_i, \mathbf{v}_j \in \mathcal{V}} -\omega |d_{i,j} - p_{i,j}| - (1 - \omega) q_{i,j} \tag{29}$$

where $\omega$ is a hyperparameter to balance the spring and damper components. This reward encoding the spring-damper model is used together with the classic rewards defined for reaching the goal position and obstacle avoidance for training ReDiG. For navigation, the reward function dynamically rewards robots based on their orientation and proximity to the goal, facilitating efficient navigation within the environment. For avoiding collisions, the reward function gives a numerical penalty to discourage robots from colliding with an obstacle or another robot. The classic reward is computed as follows:

$$R_i^{collision} = \begin{cases} \mathbf{q}_i \cdot \frac{\vec{r}_{\text{goal}}}{\|\vec{r}_{\text{goal}}\|} \times \|\mathbf{q}_i\| & \text{if } \|\vec{r}_{\text{goal}}\| > 0.0, \\ 15.0 & \text{if } \|\mathbf{p}_i - \mathbf{g}_i\|_2 < 0.1, \\ -1.5 & \text{if } \|\mathbf{p}_i - \mathbf{p}_j\|_2 < 0.2 \end{cases} \tag{30}$$

where $\vec{r}_{\text{goal}}$ is the reward vector pointing towards the goal, representing the direction and magnitude of the reward based on the robot's current position relative to the goal, and the dot product $\mathbf{q}_i \cdot \frac{\vec{r}_{\text{goal}}}{\|\vec{r}_{\text{goal}}\|}$ computes the alignment of the robot's velocity vector with the direction to the goal, scaled by the velocity magnitude to reward faster movement towards the goal.

### F.4 CHOICE OF BASELINE METHODS

We selected baselines based on both formation control capability and methodological alignment. From the algorithmic perspective, we compared against DGNN and AFOR, both of which are online multi-robot reinforcement learning methods trained with PPO. We also included the classical Leader-Follower approach, given its foundational role and continued relevance in formation control. From the capability perspective, DGNN does not support formation adaptation, whereas AFOR incorporates a spring-damper model to enable adaptive formation control.

### F.5 EXAMPLES OF COMPUTING THE CFI EVALUATION METRIC

To evaluate adaptive formation adaptation, we introduce Contextual Formation Integrity (**CFI**) metric in our paper, which is mathematically defined as:

$$w \left(1 - \delta^{-1} \min\left(|r - (\xi + \delta)|, |r - (\xi - \delta)|\right)\right) + (1 - w)\tau$$

where the first term assesses the team's efficiency in utilizing the corridor gap, where $r$ is the robot team's maximum radius, $\xi$ denotes a threshold which is the corridor width with a safety margin, and $\delta$ is an uncertainty with smaller values imposing stricter formation requirements. CFI's second term $\tau \in [0, 1]$ evaluates the integrity of the team shape. CFI combines these two terms to evaluate how effectively a robot team uses the corridor space and maintains its formation, with the balance determined by the coefficient $w$. The metric CFI $\in [0, 1]$, where higher values indicate better performance. In our experiments, we set $w = 0.5$ to treat the gap usage and the formation integrity equally important. Additionally, we set $\delta$ to twice the width of the robot used in the corresponding experiments. For a number of $n$ robots, the $\tau$ in CFI is computed as follows:

- Circle formation: $\tau = 1 - \frac{1}{n} \sum_{i=1}^{n} \frac{\theta_i}{\frac{(n-2) \times 180}{n}}$, where $\theta_i$ represents the interior angle of the triangle with the $i$-th robot as the vertex, and $\frac{(n-2) \times 180}{n}$ is the interior angle of the polygon, approximating a circle when the team has $n$ robots.

- Wedge Formation: $\tau = 1 - \frac{2|L_l - L_r|}{L_l + L_r} - \frac{|2L_m - L_b|}{L_b}$, where $L_l, L_r, L_m, L_b$ represent the lengths of the left, right, middle, and base sides of the isosceles triangle formed by the robots.

- Line Formation: $\tau = 1 - \frac{1}{n-1} \sum_{i=1}^{n-1} \frac{L_{i,i+1}}{L}$, where $L_{i,i+1}$ represents the distance between neighboring robots, and $L$ denotes the full width of the robot team. The term $\tau$ measures the relative deviation from the ideal line formation.

### F.6 AUTONOMY STACK ARCHITECTURE

We design autonomy stacks for three different platforms, including Gazebo simulation, Unity simulation, and physical robots, as illustrated in Figure 8. In all cases, each robot is assigned a unique namespace to support decentralized execution, which ensures that every robot runs its own autonomy stack, independently processing its local state and executing actions.

In Gazebo simulation, we use Ubuntu 22.04 with ROS2 Humble. The simulator provides a 2D occupancy map together with ground-truth robot states, including positions, velocities, and obstacle proximity. Goal positions are specified for each robot under its namespace. At every timestep, ReDiG generates velocity commands in the form of linear values, which are then executed by Limo robots through the `/cmd_vel` topic, where the linear command is translated into both linear and angular velocities for differential drive control.

Unity simulations are run on Ubuntu 20.04 with ROS1 Noetic. The Unity engine supplies the 2D occupancy map, robot poses, and obstacle proximity. After ReDiG computes velocity commands, Warthog robots execute them through the `/cmd_vel` topic under their respective namespaces. This stack provides a high-fidelity testing environment that replicates large-scale outdoor settings with accurate robot dynamics.

For physical robots, we use Ubuntu 22.04 with ROS2 Humble. Robot states are estimated through SLAM-based methods. Each Limo robot receives its state and goal under a dedicated namespace. ReDiG outputs single-step velocity commands that are executed in a decentralized manner through the `/cmd_vel` topic.

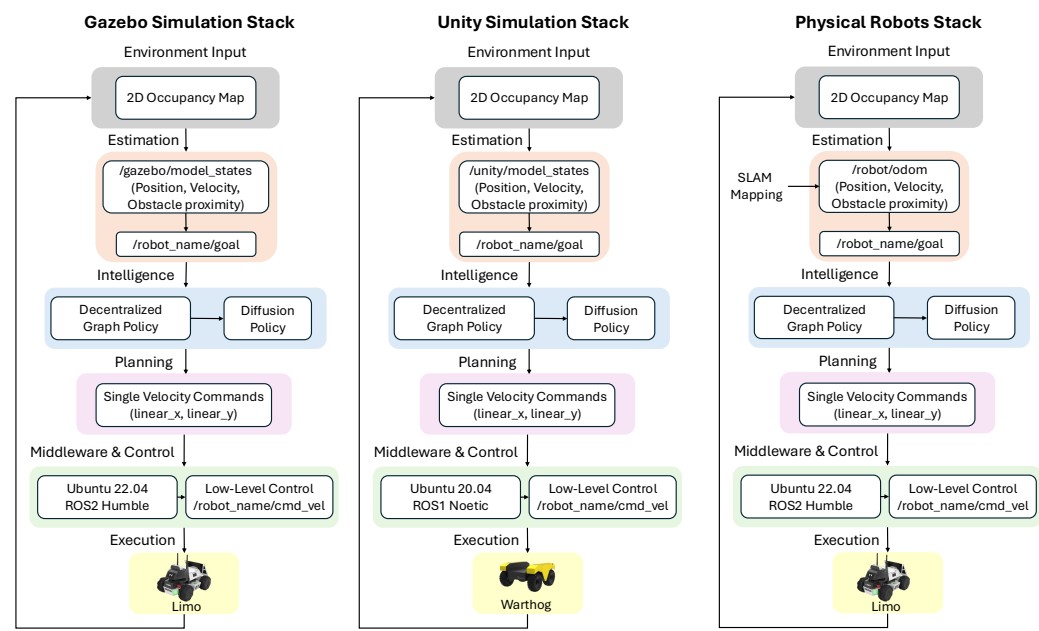

Figure 8: Autonomy stacks for robotics simulations and physical robots.

### F.7 RUNTIME AND CONTROL FREQUENCY

ReDiG generates only the current velocity vector at each timestep for each individual robot, with one full diffusion pass corresponding to a single control action. This design eliminates the need to generate full trajectories, significantly reducing latency. In simulation, we employ a workstation equipped with an NVIDIA RTX 4090 GPU and run the full diffusion process with 100 denoising steps, achieving a control frequency of approximately 128.6 Hz during rollouts.

We also observe that using early stopping (e.g., 60 steps) can increase the frequency to around 223.0 Hz without compromising performance. For real-world deployment, both the diffusion model and the GNN are executed onboard each LIMO robot using an Intel NCU i7 processor. With early stopping, higher control frequencies can be achieved when required, ensuring real-time operation under hardware constraints.

## G EXTENDED EXPERIMENTAL RESULTS

We present comprehensive qualitative experimental results to demonstrate the applicability and generalizability of ReDiG. Evaluations are conducted across diverse environments, including Gazebo with ROS2, a high-fidelity 3D Unity simulator with ROS1, and physical multi-robot teams with ROS2 in both indoor and outdoor settings.

Figure 9 shows the robotic platforms used in our experiments, where we employ ClearPath Warthog robots in simulation and Agilex Limo robots in real-world experiments. Since ReDiG directly generates linear velocity commands for each individual robot, it can be readily applied to any velocity-controlled ground robot, regardless of platform or deployment environment.

Figure 10 shows Warthog robots in the Unity simulator, where they are used to validate ReDiG's effectiveness in unstructured, high-fidelity environments. Figure 11 and Figure 12 show real-world experiments with varying numbers of Limo robots in indoor and outdoor environments, respectively, demonstrating ReDiG's adaptability to diverse operating conditions and team sizes.

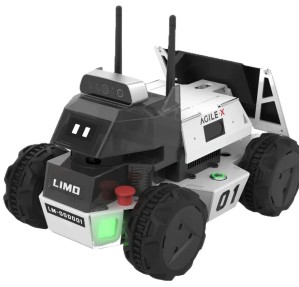
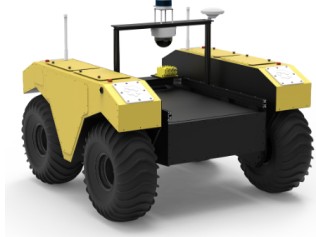

(a) Agilex Limo Robots       (b) ClearPath Warthog Robots

Figure 9: Robotic platforms used in our experiments. Figure 9(a) shows differential-drive ground robots Agilex Limo robots that are used in our real-world experiment. Figure 9(b) Clearpath Warthog robots are large, rugged, all-terrain differential-drive UGVs that are used in our Unity simulations.

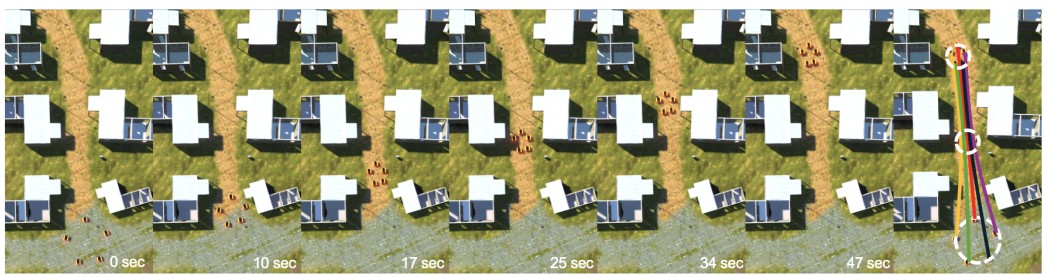

(a) A team of five robots in a circle formation navigates through multiple narrow passages.

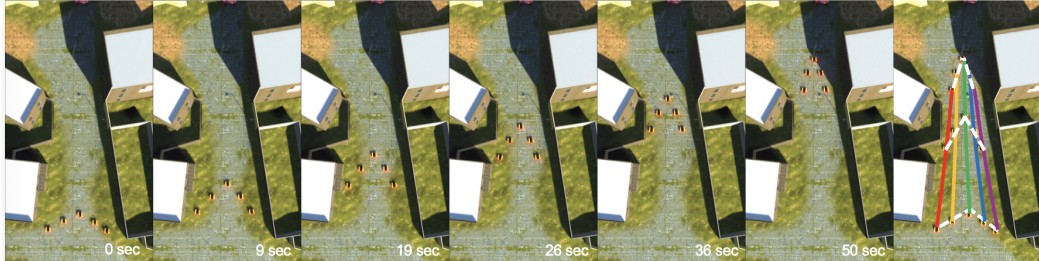

(b) A team of five robots in a wedge formation traverses a progressively narrowing corridor between buildings.

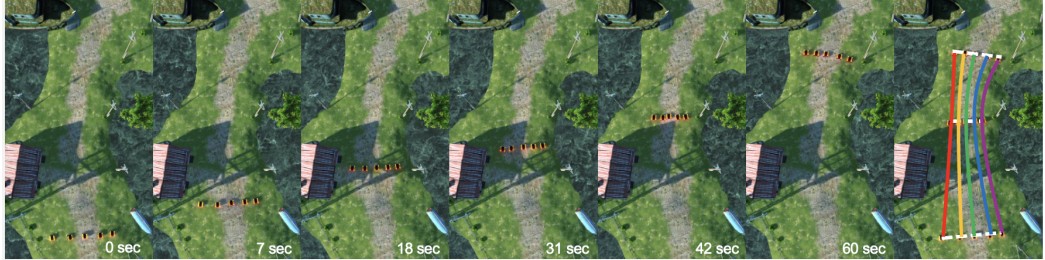

(c) A team of five Warthog robots with a line formation navigates through a narrow corridor on uneven terrain.

Figure 10: Qualitative results on ReDiG for decentralized coordinated multi-robot navigation using a high-fidelity Unity3D simulations in ROS1. The experiments utilize differential-drive Warthog robots that maintain circle, wedge and line formations while traversing an unstructured outdoor field environment.

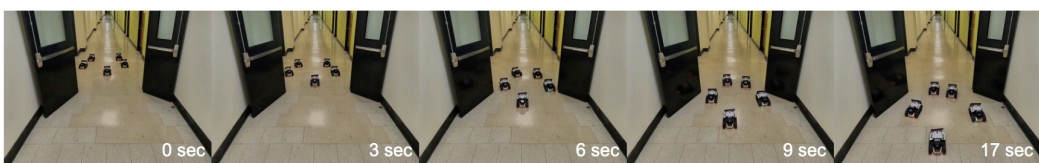

(a) A team of five physical robots with a circular formation navigates through a narrow doorway in a hallway.

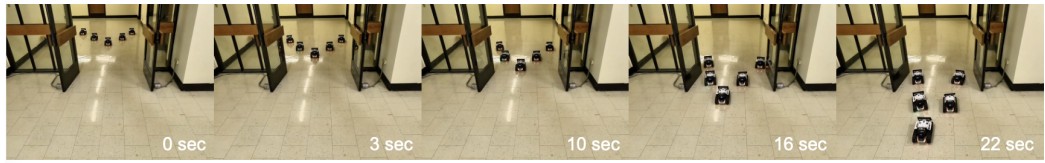

(b) A team of five robots with a wedge formation navigates through a narrow exit.

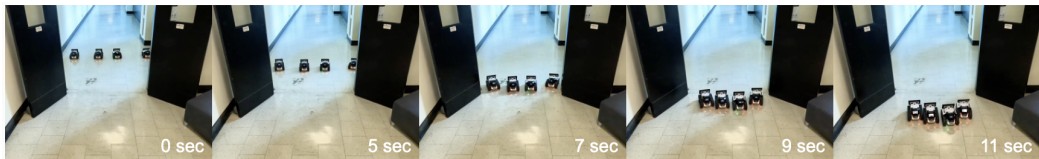

(c) A team of four robots in a line formation navigates through a narrow door.

Figure 11: Qualitative results on smooth formation adaptation are demonstrated across diverse indoor environments, with varying formation shapes and different team sizes. Using multiple differential-drive Limo robots, the teams successfully maintain circle, wedge, and line formations under a range of indoor scenarios.

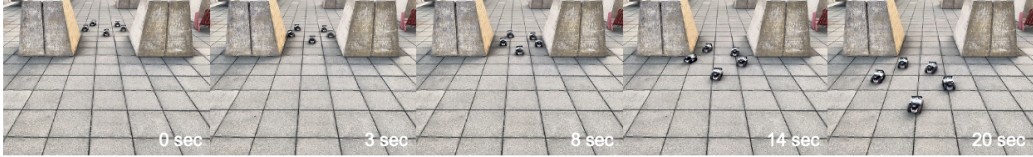

(a) A team of five physical robots with a circular formation navigates through a narrow passage between two concrete security bollards.

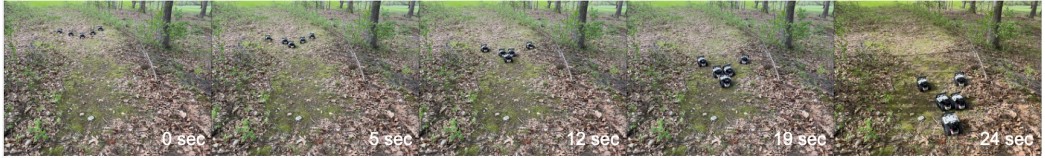

(b) A team of five robots with a wedge formation navigates through a forest-like environment characterized by narrow corridors, scattered trees, and surrounding obstacles.

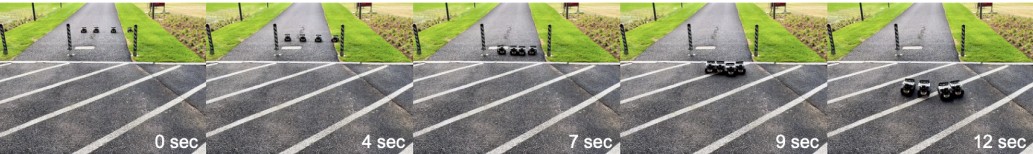

(c) A team of four robots in a line formation navigates through a narrow pathway, where robots access is restricted by two sticks marking the boundaries.

Figure 12: Qualitative results of ReDiG demonstrate decentralized coordinated navigation with smooth adaptive formation control, using different numbers of differential-drive Limo robots that maintain different formations across various unstructured outdoor environments.

## H EXTENDED RELATED WORK

**Coordinated Multi-Robot Navigation.** Traditional coordinated navigation with formation control often relies on manually designed strategies, such as the leader-follower structure Reily et al. (2020); Wu et al. (2022); Xiao & Chen (2019), where follower agents are programmed to maintain formation by tracking a designated leader. Virtual region methods Roy et al. (2018); Abujabal et al. (2023); Roy et al. (2019); Alonso-Mora et al. (2019) allow teams to adjust their formation within predefined spatial constraints. However, these formations are often rigid and lack the ability to adapt based on environments. For learning methods, Graph Neural Networks (GNNs) have been introduced to improve team coordination and communication in a decentralized manner Goarin & Loianno (2024); Zhang et al. (2023); Li et al. (2020b); Gao et al. (2024). Online RL has been widely applied in multi-agent systems Blumenkamp et al. (2022); Hu et al. (2023); Han et al. (2020); Hacene & Mendil (2021), which enables robot teams to learn complex, coordinated behaviors that are difficult to manually design. Recently, adaptive formation control has been explored by integrating online reinforcement learning with a spring–damper model, which enables balanced coordination between robot pairs Deng et al. (2025a;b). Despite these advantages, for coordinated navigation with formation control that requires synchronization between robots, RL often suffers from step-wise decision-making, which can lead robots to frequently stop or adjust their motion to maximize immediate rewards, resulting in jerky trajectories and reduced motion smoothness.

**Diffusion Models for Robot Policy Learning.** Diffusion models have gained significant attention in robotics for generating smooth trajectories through iterative denoising. For single-robot planning, diffusion models are used to sample motion plans conditioned on task objectives Carvalho et al. (1916); Ma et al. (2024) and environmental context Fang et al. (2024); Xian & Gkanatsios (2023); Chi et al. (2023); Kapelyukh et al. (2023). Hierarchical diffusion models have been proposed to handle long-horizon planning problems Li et al. (2023); Chen et al. (2024). Recent works extend diffusion models to multi-robot systems to enable coordinated trajectory generation. Motion Diffuser Jiang et al. (2023) enables trajectory prediction for multi-robot through cost function, Resilient Distributed Diffusion Li et al. (2020a) enables resilient distributed control under adversarial conditions based on the centerpoint concept, MMD Shaoul et al. (2024) generates collision-free multi-robot trajectories based on single-robot data. RDT Liu et al. (2024) enables bimanual manipulation from text commands by using language-conditioned diffusion, GSC Mishra et al. (2023) samples from the skill model to generate long-horizon plans . 3D diffusion policy Ze et al. (2024) integrates 3D visual representations (e.g. point clouds) into diffusion policies, improving generalization in real robot tasks. VPDD uses discrete diffusion to pretrain on large video data, then fine-tunes robot policies from fewer demonstrations He et al. (2024). However, applying diffusion models to multi-robot systems remains challenging due to the need for large-scale, well-synchronized expert demonstrations, which are difficult to obtain.

**Diffusion for Offline RL.** Diffusion models have been integrated with RL to improve policy through generative sampling guided by RL signals. In the offline RL setting, Diffusion-QL Wang et al. (2022) biases diffusion sampling toward high-value actions using Q-learning. CEP Lu et al. (2023) defines contrastive energy scores to steer denoising. SRDP Ada et al. (2024) enhances out-of-distribution (OOD) generalization by reconstructing state representations. CPQL Chen et al. (2023b) introduces consistency modeling for stable policy learning. Diffuser Janner et al. (2022) applies reward signals at the trajectory level, while Simple Hierarchical Chen et al. (2024) extends this to multi-task settings using hierarchical diffusion policies. MetaDiffuser Ni et al. (2023) demonstrates that incorporating conditional diffusion models into task inference significantly outperforms previous meta-RL methods. AdaptDiffuser Liang et al. (2023) enhances diffusion models with evolutionary planning to improve offline RL performance and generalization to unseen tasks. LCD Zheng et al. (2022) employs a truncated diffusion process with a hierarchical structure, enabling efficient long-horizon multi-task control while reducing the computational cost of training and generation. DoF Li et al. (2025) introduces a diffusion factorization framework for offline multi-agent reinforcement learning, enforcing the Individual-Global-Identically-Distributed principle to improve scalability and cooperation. MTDiff He et al. (2023) further supports multi-task planning through transformer-based conditioning. MADiff Zhu et al. (2024) is the first offline diffusion-based multi-agent framework. However, for complex behaviors that demand coordination and synchronization, which are rarely

available in offline datasets, offline RL struggles to learn behaviors that are absent from expert demonstrations.

**Diffusion for Online RL.** Diffusion-based online RL addresses the limitation of offline RL by directly interacting with the environment, enabling the model to explore and refine behaviors beyond those available in expert demonstrations. DIPO Yang et al. (2023) is the first to integrate diffusion policies into online RL and introduces a novel diffusion policy improvement method, which uses off-policy to refine actions through gradient ascent updates to obtain higher rewards. CPQL Chen et al. (2023a) conducts experiments showing that one-step consistency models can naturally serve as online RL policies, achieving a strong balance between exploration and exploitation. QSM Psenka et al. (2023) aligns the diffusion model's score function with the gradient of a Q-function, effectively connecting the denoising process to action-value learning. This allows policy updates to be performed by differentiating only through the denoising model, yielding multi-modal and explorative behaviors in continuous domains. QVPO Ding et al. (2024) introduces a Q-weighted variational loss to ensure robust policy improvement by tightly approximating the policy objective. This formulation adapts diffusion policies to online RL, leverages their multimodality for enhanced exploration, and incorporates entropy regularization and efficient action selection to reduce variance and improve sample efficiency. However, none of these diffusion-based online RL methods have been applied to multi-robot systems, particularly those requiring multi-robot coordination and synchronization.

