# OpenReview forum: "ReDiG: Reinforced Diffusion on Graphs for Decentralized Coordinated Multi-Robot Navigation with Smooth Formation Adaptation"
_ICLR.cc/2026/Conference — ICLR 2026 Conference Withdrawn Submission_

### Official Review · Reviewer_t7Md · 2025-10-14

**Soundness:** 2
**Presentation:** 3
**Contribution:** 2
**Rating:** 4
**Confidence:** 4

**Summary:**

This paper develops Reinforced Diffusion on Graphs (ReDiG), a unified learning paradigm for multi-robot navigation. ReDiG contains three components: (1) a graph neural network for decentralized coordination, (2) a diffusion model for individual trajectory generation, and (3) an online reinforcement learning method for team synchronization. Experiments conducted in both simulated and real-world scenarios with various settings illustrate the remarkable performance of the proposed approach.

**Strengths:**

1. The paper is well-structured and presented in a clear academic style.
2. The authors provide strong motivation for their approach.
3. Experiments with various robot types and formation shapes validate the effectiveness of ReDiG.

**Weaknesses:**

1. ReDiG is a combination of several different algorithms rather than an elegant one.
2. The graph neural network (GNN) is used as an encoder to encode the robot's observation and state for diffusion models. Similar techniques have been widely employed in prior works [1, 2].
3. A standard diffusion model is used to generate trajectories without any guidance or projection, and prior work indicates its performance degrades rapidly as the number of robots increases [3, 4].
4. The number of robots used in experiments is very limited (even in simulated environments). There is no challenging scenario considered in experiments.
5. The Contextual Formation Integrity (CFI) of ReDiG is lower than that of AFOR in multiple settings.
6. No significance testing is reported across methods





**References**:

[1] Wang, Yutong, et al. "Scrimp: Scalable communication for reinforcement-and imitation-learning-based multi-agent pathfinding." 2023 IEEE/RSJ International Conference on Intelligent Robots and Systems (IROS). IEEE, 2023.

[2] Ma, Yixiao, et al. "Privileged Reinforcement and Communication Learning for Distributed, Bandwidth-limited Multi-robot Exploration." arXiv preprint arXiv:2407.20203 (2024).

[3] Shaoul, Yorai, et al. "Multi-Robot Motion Planning with Diffusion Models." The Thirteenth International Conference on Learning Representations.

[4] Liang, Jinhao, et al. "Simultaneous Multi-Robot Motion Planning with Projected Diffusion Models." Forty-second International Conference on Machine Learning.

**Questions:**

1. The authors claim that ReDiG can generate smooth trajectories. Did they employ any guidance or additional methods during the training or sampling process to ensure the smoothness of the generated trajectories, or did they simply rely on a standard diffusion model?
2. Does the proposed approach provide any theoretical guarantees regarding feasibility, smoothness, or formation?
3. What are the smoothness results of the baseline methods?

---

> ### Author Response · Authors · 2025-11-21
>
> We thank the reviewer for your constructive feedback. Please find our detailed responses to each of your comments below.
>
> >**Q1:** ReDiG is a combination of several different algorithms rather than an elegant one.
>
> Please refer to our response to Q2 of Reviewer qZAD, which clarifies the novelty of our work. As a sketch, our contributions include the online diffusion-based multi-robot learning framework, a RL refinement mechanism, and a GNN policy for decentralized and scalable coordination.
>
> >**Q2:** The number of robots used in experiments is very limited (even in simulated environments). There is no challenging scenario considered in experiments.
>
> We agree that the number of robots used in our experiments is limited to fewer than five. However, we provide a detailed scalability and time-complexity analysis of our algorithm, please refer to our response to Q3 of Reviewer qZAD.
> In addition, all experimental environments include narrow corridors that are infeasible for rigid formation control and therefore require adaptive formation behaviors. The Unity simulation environments mimic outdoor scenarios with extended narrow pathways that require long, curved trajectories and continuous formation adaptation. The real-world experiments include both indoor hallway environments and outdoor forest-like environments.
>
> >**Q3:** The Contextual Formation Integrity (CFI) of ReDiG is lower than that of AFOR in multiple settings.
>
> We agree that ReDiG exhibits slightly lower CFI scores than AFOR in several settings. This represents a reasonable trade-off for the substantial gains in efficiency and responsiveness, as faster movement can sacrifice strict formation maintenance. However, this is not a fixed limitation. The balance between formation adaptation and navigation objectives in our reward function is controlled by a tunable hyperparameter, which enables ReDiG to flexibly trade off between formation integrity and execution efficiency depending on task requirements.
>
> >**Q4:** The authors claim that ReDiG can generate smooth trajectories. Did they employ any guidance or additional methods during the training or sampling process to ensure the smoothness of the generated trajectories, or did they simply rely on a standard diffusion model?
>
> We do not employ any additional guidance beyond a standard diffusion formulation. First, the denoising network in the diffusion model learns to remove high-frequency perturbations introduced by Gaussian noise, guided by the reverse process coefficients, which promotes smooth trajectory generation.
> Second, the spring-damper model reward function penalizes sudden relative motion between robots, further encouraging smooth and stable movement. While AFOR employs the spring-damper model, its step-wise decision-making nature of RL can lead to frequent adjustment and oscillated movement. In contrast, ReDiG generate actions through a diffusion process, which mitigates this issue and results in smoother trajectories.
>
> >**Q5:** Does the proposed approach provide any theoretical guarantees regarding feasibility, smoothness, or formation?
>
> We provide theoretical guarantees for feasible formation adaptation in Theorems 1 and 2. The synchronization of formation adaptation among robots is achieved through the spring–damper reward function. In particular, given this formation adaptation reward, Theorem 2 establishes the convergence of the critic, ensuring that it provides reliable Q-value estimates for actions generated by the diffusion model while explicitly accounting for formation adaptation objectives. Using these critic-refined actions as new supervision targets, the diffusion model performs denoising updates. Theorem 1 then guarantees the convergence of the diffusion process to actions that consider formation adaptation during navigation. Together, these results provide a theoretical guarantee that ReDiG can achieve feasible formation adaptation.
>
> >**Q6:** What are the smoothness results of the baseline methods?
>
> We analyze motion trajectory smoothness in Section 5 and Figure 7 using jerk as our evaluation metric. DGNN, which does not incorporate diffusion or a spring-damper model, exhibits the most jerky trajectories. AFOR, which includes a spring–damper model but does not use diffusion for individual robot navigation, achieves smoother motion but still suffers from noticeable oscillations. In contrast, ReDiG integrates both a diffusion model and a spring–damper model, which achieves the lowest jerk and therefore the smoothest trajectories among all compared methods.

---

> > ### Comment · Reviewer_t7Md · 2025-11-23
> >
> > Thanks for the response! However, two significant concerns remain unsolved:
> >
> > > **Empirical Scalability**
> > Authors cannot provide empirical evidence of the scalability of their methods. While I appreciate the time-complexity analysis, it is different from the navigation success capability. In other words, the time-complexity analysis is not a guarantee of scalability in Multi-Robot Navigation.
> >
> > >**Trajectory Smoothness**
> > The *Ablation Study on Motion Trajectory Smoothness* in Section 5 does not provide the smoothness results of the baseline methods, such as L&F and DGNN.

---

> > > ### Author Response · Authors · 2025-11-24
> > >
> > > We thank the reviewer for the valuable feedback.
> > > >**Q7:** Empirical Scalability.
> > >
> > > We thank the reviewer for this important point. We agree that time‐complexity analysis is not a direct guarantee of navigation success, and that large‐scale empirical evaluation is an important complementary validation.
> > > However, scalability and navigation success characterize two fundamentally different aspects of a multi‐robot system.
> > > Navigation success measures task‐level performance (e.g., goal completion), whereas scalability is primarily constrained by how computational costs grow as the number of robots increases. In this context, time‐complexity analysis provides a formal, algorithmic indicator of scalability by explicitly characterizing the growth rate of computation with respect to team size. Without favorable complexity, an approach cannot be executed in real time for large robot teams, regardless of its theoretical navigation capability. Therefore, while empirical success rates reflect performance, our time‐complexity analysis establishes a necessary and widely used notion of scalability in multi‐robot planning and control.
> > >
> > >
> > > >**Q8:** Trajectory Smoothness.
> > >
> > > | Method  |Mean Jerk Averaged Across Robots|
> > > |:--------|:--------:|
> > > | L & F|  0.250722   |
> > > | DGNN |  0.442648   |
> > > | AFOR |   0.171200   |
> > > | ReDiG (ours)|   0.080129 |
> > >
> > > We evaluate the motion smoothness of all baseline methods in the wedge formation using the jerk metric. The Mean Jerk Averaged Across Robots is computed by averaging the jerk over time for each robot and then averaging across the entire team to obtain a global smoothness score. As shown in the table, our method achieves the smoothest trajectories during navigation with formation adaptation.

---

> > > > ### Comment · Reviewer_t7Md · 2025-11-24
> > > >
> > > > Thanks for the response!
> > > >
> > > > My point is that time‐complexity analysis alone is not enough to claim scalability. In the context of multi-robot systems, scalability usually refers to the ability to maintain a high success rate as the number of robots increases. Specifically, an algorithm with excellent time complexity but fails in its goals does not make sense. I am open to changing my score if authors can provide competitive empirical results in larger systems (e.g., >10 robots).

---

### Official Review · Reviewer_AVUS · 2025-10-28

**Soundness:** 3
**Presentation:** 4
**Contribution:** 3
**Rating:** 6
**Confidence:** 4

**Summary:**

ReDiG presents a decentralized online learning framework that synergizes graph neural networks, conditional diffusion policies, and actor-critic reinforcement learning to achieve smooth, adaptive formation navigation for multi-robot teams. Theoretically grounded by convergence bounds for both diffusion and value approximation, the method is validated in simulation and on physical robots, demonstrating great task success rate and superior trajectory smoothness. Limitations include shape-specific retraining, reliance on single-robot demonstrations.

**Strengths:**

1.Propose a new multi-robot collaboration framework for formation controlling. This framework unifies decentralized graph neural networks, diffusion-based trajectory generation, and reinforcement-learning-driven formation synchronization.

2.Rigorous convergence guarantee: Proves the explicit upper bounds on the KL divergence between learned and true action distributions for the conditional diffusion model, isolating prior mismatch, denoising error, and discretization error; provides a parallel finite-sample bound for the critic that separates statistical and algorithmic errors, ensuring monotonic improvement of both trajectory smoothness and value estimates during online training.

3.Eliminates expert-demonstration bottleneck : This article bootstraps a diffusion policy via combining it with online RL, converting environmental reward into synchronized, formation-aware demonstrations without costly multi-robot experts.

**Weaknesses:**

1.Fixed communication radius: The topology of multi-robot system is set manually, the ability of adaptation is absent.

2.Formation-specific training: A separate model must be retrained for each desired shape (wedge, line, circle), precluding on-the-fly formation switching.

3.Demonstration dependency: Initial supervision relies on single-robot planners (A*, RRT); no curriculum or self-supervised pre-training is explored.

**Questions:**

1.Although 60-step early stopping achieves 223Hz, the latency and CPU/GPU utilisation for the full 100-step model are not given; such data are critical for deployment on resource-constrained situations.

2.Recent multi-agent diffusion or offline RL baselines (MADiff, DoF) are omitted; a comparative discussion would verify ReDiG’s contribution.

3.Only aggregate success rates are presented. Illustrative failure cases—e.g., localization drift, packet loss, or corridor congestion—and their recovery statistics would help ascertain robustness.

---

> ### Author Response · Authors · 2025-11-21
>
> We thank the reviewer for insightful comments and for appreciating our contribution. Our response to each key comment is provided below.
>
> >**Q1:** Although 60-step early stopping achieves 223Hz, the latency and CPU/GPU utilisation for the full 100-step model are not given; such data are critical for deployment on resource-constrained situations.
>
> We evaluated ReDiG on a machine equipped with a 16-core Intel i9 CPU, 32 GB of RAM, and an NVIDIA RTX 4090 GPU. When running the full diffusion process with 100 denoising steps, the controller achieves an average frequency of approximately 128.6 Hz during rollouts. This corresponds to approximately 0.4 \% CPU utilization, 5.4 \% GPU utilization, and 7.4 \% GPU memory usage, which indicates substantial computational headroom. These results suggest that ReDiG can be deployed efficiently on larger multi-robot systems and is well-suited for resource-constrained onboard hardware.
>
> >**Q2:** Recent multi-agent diffusion or offline RL baselines (MADiff, DoF) are omitted.
>
> First, our method uses diffusion-based online reinforcement learning, where robots interact with the environment and learn through it. In contrast, MADIFF and the referenced works are offline approaches that can only replicate behaviors present in the dataset. While we are aware of and have tested these offline methods, we found that they cannot learn complex behaviors like formation adaptation, which require coordination and synchronization but are rarely available in offline data. Our method addresses this by enabling learning of such behaviors through online interaction. Second, MADIFF generates state trajectories and relies on an inverse dynamics model to infer actions, whereas our approach directly generates actions at each timestep for continuous control. This enables better real-time execution and reduces error propagation introduced by decoupling the state generation and action inference processes.
>
> >**Q3:** Only aggregate success rates are presented. Illustrative failure cases—e.g., localization drift, packet loss, or corridor congestion—and their recovery statistics would help ascertain robustness.
>
> While we did not include illustrative failure cases in the main paper, we have observed several representative failure cases during testing.
> First, ReDiG currently relies on external state estimation. Inaccurate pose or velocity estimates can lead to unexpected motion, such as oscillation.
> Second, ReDiG is trained in a 2D environment on flat surfaces. When deployed in more complex 3D environments (e.g., uneven terrain), discrepancies between the training assumptions and real-world dynamics can affect velocity tracking and formation consistency.
>
> >**Q4:** Fixed communication radius: The topology of multi-robot system is set manually, the ability of adaptation is absent.
>
> While a predefined communication radius is used to construct the interaction graph, ReDiG is not limited to a fixed topology in principle. The graph is dynamically constructed at every timestep based on the current robot states, and edges only formed between robots within the specified range. This radius is a tunable hyperparameter, not a constraint, and can be made adaptive (e.g., density-based) without changes to the overall architecture of our framework.
>
> >**Q5:** Formation-specific training: A separate model must be retrained for each desired shape (wedge, line, circle), precluding on-the-fly formation switching.
>
> We agree that separate models are currently trained for each formation type to enable controlled evaluation. Although we did not explicitly analyze on-the-fly formation switching in this work, we regard it as an important and non-trivial challenge, and treat it as part of our planned future work. At present, formation switching can be realized by replacing the diffusion policy at runtime. A more principled solution is to introduce an additional decision layer that selects the appropriate formation based on the perceived environmental context.

---

### Official Review · Reviewer_qZAD · 2025-10-30

**Soundness:** 2
**Presentation:** 3
**Contribution:** 2
**Rating:** 2
**Confidence:** 3

**Summary:**

This paper presents ReDiG, a framework which integrates GNNs, diffusion models, and RL to enable multi-robot motion planning. The method relies on a graph neural network to coordinate between trajectories generated by independent diffusion models corresponding to each robot. This ensemble is well motivated by prior success utilizing GNNs, and the reported results demonstrate greater efficiency than the compared baselines.

**Strengths:**

- **Technical Presentation:** The paper is written in a technical format that is appropriate for the target audience. The exposition is fairly intuitive.

- **Real-World Evaluation:** The inclusion of results from real-world deployment are compelling, providing support for the authors' claims of practical utility.

- **Time Travel Efficiency:** The faster paths provided by ReDiG provide are interesting, indicative of more optimal paths.

**Weaknesses:**

- **Baseline Selection:** It is surprising that the paper does not compare to any recent diffusion motion planning baselines, e.g., [1-3]. This seems appropriate, especially when claiming SOTA performance. It is difficult to assess the performance on the method without this analysis. Comparison is limited to older RL baselines and GNN-based approaches; because of this, the evaluation seems incomplete. Could the authors please comment on why these methods have not bee compared to?

- **Scope of Contribution:** Presently, I'm not convinced of the overall novelty of the framework. While this ensemble of methods is unique, the methodology centers on ensembling existing tools (e.g., GNNs for coordination, Diffusion Models for motion planning, and RL). While this is indeed a contribution, it seems to fall more on the engineering side of things.


---

[1] Carvalho, Joao, et al. "Motion planning diffusion: Learning and planning of robot motions with diffusion models." 2023 IEEE/RSJ International Conference on Intelligent Robots and Systems (IROS). IEEE, 2023.

[2] Shaoul, Yorai, et al. "Multi-robot motion planning with diffusion models." arXiv preprint arXiv:2410.03072 (2024).

[3] Liang, Jinhao, et al. "Simultaneous Multi-Robot Motion Planning with Projected Diffusion Models." arXiv preprint arXiv:2502.03607 (2025).

**Questions:**

- It seems that better TT comes at the cost of tolerance to $\delta$? Is ReDiG capable of performing this trade-off (e.g., tighter tolerance is required for a particular task)? Has any analysis of this been conducted?

- Could the authors clarify which components of the framework are algorithmically novel, as opposed to a composition of existing techniques?

- What limitations currently prevent ReDiG from scaling to a higher number of robots?

---

> ### Author Response · Authors · 2025-11-21
>
> We thank the reviewer for the constructive feedback. Our response to each of the comment is provided below.
> >**Q1:** Comparision with multi-robot diffusion-based baselines.
>
> First, the referenced diffusion-based methods focus on offline training, which makes them unsuitable for a fair comparison with our online learning framework. While we are aware of and have tested these offline methods, we found that they cannot learn complex behaviors like formation adaptation, which require coordination and synchronization but are rarely available in offline data. Our method addresses this by enabling learning of such behaviors through online interaction.
> For this reason, we prioritized comparisons with state-of-the-art online RL methods explicitly designed for multi-robot coordination and formation control, making them more appropriate for meaningful evaluation, such as AFOR (ICRA'25) and DGNN (ICRA'22).
>
> >**Q2:** Could the authors clarify which components of the framework are algorithmically novel.
>
> Our framework introduces several algorithmic contributions beyond a simple combination of existing techniques. We present it as the first online diffusion-based multi-robot learning framework. Most existing diffusion-based approaches for multi-robot systems operate in an offline setting and rely on large-scale, high-quality expert demonstrations. However, collecting synchronized and coordinated expert data for challenging problems, such as formation adaptation, is difficult in practice. To address this limitation, we propose an online RL diffusion-based framework that allows the diffusion model to continuously refine its policy through interaction with the environment, even when the available expert demonstrations are imperfect. Moreover, we integrate a GNN into the diffusion-based policy to enable decentralized coordination across the robot team, which improves scalability to larger robot teams.
>
> >**Q3:** What limitations currently prevent ReDiG from scaling to a higher number of robots?
>
> We analyze the scalability of ReDiG in Section 5 and Appendix E. The primary limitation arises during training, where the time complexity is dominated by $O(n^2)$ due to pairwise message passing in the GNN encoder. This cost is partially amortized through batch parallelization and remains tractable for moderately large robot teams.
> During decentralized execution, the complexity reduces to $O(k+T_d)$, where $k$ is the number of neighboring robots and $T_d$ denotes the number of diffusion denoising steps. As a result, the effective runtime is dominated by the diffusion process rather than the team size, and the per-robot computation remains bounded. These properties enable ReDiG to scale efficiently to larger robot teams in practice.
>
> >**Q4:** It seems that better TT comes at the cost of tolerance to $\delta$? Is ReDiG capable of performing this trade-off (e.g., tighter tolerance is required for a particular task)? Has any analysis of this been conducted?
>
> We use $\delta$ as an evaluation metric to indicate the strictness of the formation adaptation requirement. We agree that there is a reasonable trade-off between travel time and formation integrity, since faster motion can sacrifice strict formation maintenance.
> In ReDiG, this balance is controlled by a tunable hyperparameter in the reward function that trades off the formation adaptation term against the navigation term. By adjusting this hyperparameter, ReDiG can balance formation maintenance and navigation depending on the task requirement.

---

### Official Review · Reviewer_igfk · 2025-10-31

**Soundness:** 2
**Presentation:** 2
**Contribution:** 2
**Rating:** 2
**Confidence:** 4

**Summary:**

This paper proposes a learning-based method to achieve decentralized multi-robot navigation with smooth formation adaptation. Experimental results in both indoor and outdoor environments demonstrate the effectiveness of the proposed method.

**Strengths:**

Extensive experiments are conducted in both indoor and outdoor environments using physical robot teams and robotics simulations.

**Weaknesses:**

The related work section is insufficiently discussed. The use of the diffusion model and its training process are unclear. The proposed algorithm is not clearly described and seems questionable. See Questions part for more details.

**Questions:**

(1) Since this paper focuses on the distributed formation navigation problem, the authors are strongly suggested to compare their method with existing control-based methods, such as bearing-based and angle-based formation control methods, which enable multi-robot systems to reduce their formation size while maintaining the formation shape when passing through narrow passages.

(2) It is stated in the appendix that classic path planning algorithms (e.g., RRT) can be used to generate expert demonstrations. Could you clarify how the expert trajectories are aligned or matched with the state–action pairs in your method?

(3) The loss function in (1) is hard to understand. How do you determine $\epsilon_k$? Why do you use $a_i^0$ instead of $a_i^k$? How is the notation $\psi$ related to this loss function?

(4) The gradient form of the loss function (1) in line 6 in Algorithm 1 seems wrong.

(5) The actor update step in line 11 in Algorithm 1 seems wrong.

(6) Please give the specific gradient formula for the GNN parameters.

(7) The authors put only the expert trajectory and the transition generated by the diffusion model in the replay buffer. How do you update your actor?

---

> ### Author Response · Authors · 2025-11-21
>
> We thank the reviewer for giving valuable feedback. Our response to each of the comments is provided below.
>
> >**Q1:** Related work section is insufficient discussed.
>
> Please refer to Appendix H for an extended related work review, which covers coordinated multi-robot navigation, diffusion models for robot policy learning, diffusion for offline RL, and diffusion for online RL.
>
> >**Q2:** Comparision with control-based methods.
>
> Existing control-based methods primarily focus on maintaining a rigid formation rather than enabling formation adaptation, which makes them unsuitable for a fair comparison with our approach. These methods typically struggle in challenging environments such as narrow corridors, where our method achieves significantly better performance. We included the classical Leader–Follower baseline because it remains one of the most widely used and foundational strategies in formation control. However, bearing-based and angle-based controllers rely on predefined geometric constraints and do not support dynamic adaptation of the formation size or shape, which is essential for our problem setting. As further justification, previous classic methods did not develop metrics to evaluate formation adaptation (AFOR introduced CFI in their paper, as one of the first such metrics). In addition, our approach is an online learning based framework. Therefore, we prioritized comparisons with state-of-the-art learning methods that are more closely aligned with our methodology and objectives, ensuring a fair and meaningful evaluation.
>
> >**Q3:** Could you clarify how the expert trajectories are aligned or matched with the state–action pairs in your method?
>
> Expert demonstrations are generated independently for each robot, and produce individual waypoint sequences rather than coordinated team trajectories. At each timestep, we extract the corresponding state-action pair required by ReDiG. The robot state $s_i$ includes its position, velocity, goal position, and obstacle proximity, while the action $a_i$ is defined as the linear velocity command that drives the robot toward its next waypoint. Each transition $(s_i, a_i, R_i, s_i')$ is then stored in the replay buffer $M$.
>
> >**Q4:** How do you determine $\epsilon_k$? Why do you use $a_i^0$ instead of $a_i^k$? How is the notation $\psi$ related to this loss function?
>
> First, at each diffusion step $k$, we sample $\epsilon_k \sim \mathcal{N}(0, I)$ and construct the noisy action $a_i^{k} = \sqrt{\alpha_k}a_i^{0} + \sqrt{1 - \alpha_k}\epsilon_k$.
> Second, $a_i^{0}$ is the ground-truth clean action, while $a_i^{k}$ is the noisy action at step $k$. Following the standard diffusion formulation, the denoising network $\epsilon_k$ learns to predict the added noise $\epsilon_k$.
> Third, $\psi$ denotes the complete diffusion model, $\epsilon_\theta$ is its learnable denoising network. During sampling, $\psi$ uses $\epsilon_\theta$ to construct $a_i^{k-1}$ from $a_i^{k}$. Therefore, Eq. (1) defines the training objective for $\epsilon_\theta$.
>
> >**Q5:** The gradient form of the loss function (1) in line 6 in Algorithm 1 seems wrong.
>
> Thank you for pointing this out. We have adjusted the gradient form of Eq.(1) as follows:
> $\nabla_{\theta}\mathbb{E}\big[ \| \epsilon_k - \epsilon_{\theta}(a_i^k, k \mid h_i) \|^2 \big]$
>
>
> >**Q6:** The actor update step in line 11 in Algorithm 1 seems wrong.
>
> This is a misunderstanding. Line 11 corresponds to action refinement, not actor update. In our approach, the actor is the diffusion model, and it is updated in Line 7 via the objective in Eq. (1). Line 11 performs gradient ascent on the critic to refine the sampled actions, and these refined actions are then stored as new training targets for the diffusion model. This design enables effective online off-policy learning without backpropagating through the sampling process, which is a key contribution of our online diffusion-based multi-robot learning paradigm.
>
> >**Q7:**  Please give the specific gradient formula for the GNN parameters.
>
> The GNN parameters $W^z$ and $W^h$ are updated using the gradients from both the diffusion loss and the RL loss, as stated in Section 3.2.3. Formally, the gradient is defined as $\nabla_{W^z, W^h} L=\nabla_{W^z, W^h} L_{diff}+\nabla_{W^z, W^h} L_{critic}$, where $L_{diff}$ and $L_{critic}$ are defined in Eq. (1) and Eq. (2), respectively.
>
>
> >**Q8:** The authors put only the expert trajectory and the transition generated by the diffusion model in the replay buffer. How do you update your actor?
>
> This is a misunderstanding. In our framework, the actor is the diffusion model itself, not a separate policy network. As clarified in Q6, the diffusion model is updated via the objective in Eq. (1). The replay buffer stores initialized expert transitions and diffusion-generated transitions only to support critic training and action refinement.

---

### Comment · Area_Chair_YcZP · 2025-11-24

Dear Reviewers,

The authors have responded to your reviews. Please review and respond to their comments who have not yet done so.

Best, Your AC

---

### Note · Authors · 2026-01-22

I have read and agree with the venue's withdrawal policy on behalf of myself and my co-authors.